# Olfactory modulation of barrel cortex activity during active whisking and passive whisker stimulation

Anthony Renard [1,2,3], Evan R. Harrell [1,2,4] & Brice Bathellier[1,2]✉

Rodents depend on olfaction and touch to meet many of their fundamental needs. However, the impact of simultaneous olfactory and tactile inputs on sensory representations in the cortex remains elusive. To study these interactions, we recorded large populations of barrel cortex neurons using 2-photon calcium imaging in head-fixed mice during olfactory and tactile stimulation. Here we show that odors bidirectionally alter activity in a small but significant population of barrel cortex neurons through at least two mechanisms, first by enhancing whisking, and second by a central mechanism that persists after whisking is abolished by facial nerve sectioning. Odor responses have little impact on tactile information, and they are sufficient for decoding odor identity, while behavioral parameters like whisking, sniffing, and facial movements are not odor identity-specific. Thus, barrel cortex activity encodes specific olfactory information that is not linked with odor-induced changes in behavior.

[1] Institut Pasteur, INSERM, Institut de l'Audition, 63 rue de Charenton, F-75012 Paris, France. [2] Paris-Saclay Institute of Neuroscience, UMR9197 CNRS/ University Paris-Saclay, Campus CEA, 151 Rte de la Rotonde, 91400 Saclay, France. [3]Present address: Laboratory of Sensory Processing, Brain Mind Institute, Station 19, École Polytechnique Fédérale de Lausanne, CH-1015 Lausanne, Switzerland. [4]Present address: Interdisciplinary Institute for Neuroscience (IINS), UMR CNRS 5297, Université de Bordeaux, Centre Broca Nouvelle-Aquitaine, 146 rue Leo Saignat, CS 61292 CASE 130, 33076 Bordeaux Cedex, France. ✉email: brice.bathellier@cnrs.fr

Perception occurs through the coordinated evaluation of information from multiple senses[1]. This coordination is evident when considering the well-documented cross-modal illusions in humans[2–4], such as the ventriloquist illusion, in which erroneous sound localization is generated by visual cues. Illusions reveal strong associations across particular cues from different sensory modalities that can be maintained even if the resulting perception breaks with the physical reality. Such associations are generally useful to improve sensory judgments when information from each sense is scarce or ambiguous. In line with this, multisensory interactions are strongest when unisensory ambiguity is highest[2,5]. This inverse-effectiveness phenomenon is interpreted as an optimal integration of available information[5]. Neurons were identified in the associative cortical areas of monkeys[6,7] and rodents[8,9] which sum inputs representing congruent information from distinct modalities. This simple multisensory integration mechanism can explain inverse effectiveness and most perceptual observations, but not all[10]. A causal link between multisensory integration and associative integration has also been established[8].

While neurons in the association cortex do exhibit multimodal coding properties, several studies suggest that cross-modal connections already exist in the primary sensory cortical areas both in primates[11] and rodents[12]. These connections are functional. Most recently, it was shown that direct projections from the primary auditory cortex modulate supragranular primary visual cortex neurons in a context-dependent manner[13–15]. Even if the perceptual impact of such low-level auditory–visual interactions remains a puzzle, there is convergent evidence that they sharpen and emphasize cortical representations of the visual stimuli that are coincident with startling sounds[14,15], without a particular congruence of visual and auditory cues. This suggests the existence of multi-sensory mechanisms that are complementary to the classical cross-modal integration observed at associative levels. However, so far, it is unclear whether such interactions represent a generic computational process in primary sensory cortical areas or a feature specific to audition and vision in rodents.

To investigate this question, we focused on another pair of sensory modalities. While primates and humans rely primarily on vision and audition for survival, rodents depend to a much larger degree on olfaction and whisker touch. This reliance is underscored by an exquisite coupling between sniffing and whisking rhythms[16] and by observations that rats can easily solve a task that requires the integration of olfactory and tactile cues[17]. Nevertheless, it is unknown how these two crucial modalities interact in the brain. Activity in the barrel cortex is modulated by breathing, and this modulation depends on an intact olfactory pathway[18], but there is no evidence that odor-related information arrives in the barrel cortex. To examine the impact of odors on touch processing, we performed two-photon calcium imaging in the barrel cortex of awake mice during precisely controlled olfactory and tactile stimulations. We found that both barrel cortex activity and whisking behavior are impacted by odor stimulation. However, neither abolition of whisking by facial nerve sectioning nor pharmacological blockade of local cholinergic signaling could eliminate odor-related activity in the barrel cortex. In addition, no odor-specific information could be gleaned from facial movement behavior. This indicates that it depends on olfactory-related, secondary projections into the somatosensory system which do not include the cholinergic attentional system. In support of this, odor identity could be readily decoded from barrel cortex population activity, but the presence of olfactory information did not impact the quality of tactile representations. Therefore, our study reveals that, in the absence of learned associations, barrel cortex activity contains specific olfactory information encoded in a subspace of the neural representation.

## Results

### Calcium imaging of L2/3 barrel cortex neurons during olfacto-tactile stimulation.

The first difficulty in investigating potential cross-talk between olfactory and tactile processing is to precisely control coincident olfactory and tactile stimulations. To achieve this, we coupled a motorized tactile-object-presenting wheel with a custom-made olfactometer to synchronously present oriented tactile gratings (0° or 90°) and odors (amyl acetate or ethyl butyrate diluted at 0.1%; Fig. 1a) to the snout of a mouse. The olfactometer was calibrated with a photoionization detector (PID; Supplementary Fig. 1). Awake mice were head-fixed with their nose confined in a constant and isolated air stream to prevent the airflow from causing movements of the whiskers. In dark conditions with infrared backlighting, oriented tactile gratings were brought within range of the whiskers using a linear stage. Whisker interactions with the gratings were recorded with a high-speed infrared camera and breathing was monitored with a pressure sensor placed perpendicular to the odor stream (Fig. 1d).

We considered two stimulation contexts (Fig. 1b): in the first one, the active context, the animal was free to explore a grating that came into reach of its whiskers; in the second one, the passive context, the inferior and superior buccal branches of the facial nerve were sectioned bilaterally, abolishing whisking, and the tactile stimulation consisted of a single back-and-forth sweep of the grating against the whiskers. In both stimulation contexts, the onset of odor presentation to the nose was precisely synchronized with the time when the grating reached its fixed position (active context) or at first possible contact with the whiskers (passive context).

To record large populations of barrel cortex neurons during presentation of odors and tactile gratings, we performed stereotactic injections of AAV1-syn-GCaMP6s at 3–4 locations in the barrel cortex centered around the C2 barrel (AP—1.6, ML—3.3, DV—0.5)[19]. After ~4 weeks, location of the GCaMP6s expression locus in the barrel cortex was validated using intrinsic imaging (Fig. 1c), and mice were placed under a two-photon microscope for calcium imaging of large populations of supragranular neurons (imaging depth from 100 μm to 250 μm; Fig. 1d). One or two populations at different cortical depths were imaged per mouse in the active context and 1–4 locations in the passive context. A total of 9714 neurons were recorded in the active whisking context from 20 sessions with 12 mice; and 14,408 neurons from 19 sessions with 6 mice in the passive context. Among those populations, 1907 (19.9%) and 5162 (38.5%) neurons from the active and passive contexts, respectively, were responsive to at least one of the nine stimulation conditions (Supplementary Fig. 2a; significance threshold = 5%; Kruskal–Wallis test) and were kept for analysis. Among these populations of responsive cells, 49.4% and 68.9% responded only to touch, while 12.5% and 3.8% responded only to odors, and 5.8% and 7% responded to both touch alone and odors alone in the active and passive contexts, respectively; suggesting the presence of odor-related responses (Supplementary Fig. 2b; significance threshold = 5%; Mann–Whitney $U$ test). Calcium traces from four example neurons illustrate responses to the nine unique stimulus conditions, which include uni- and bimodal stimulus conditions as well as a blank (Fig. 1e), and show that odors can evoke responses in barrel cortex cells (neurons 1 and 2), or can modulate responses to tactile gratings (neurons 3 and 4).

### Odors modulate responses in L2/3 barrel cortex in freely whisking, head-fixed mice.

We first analyzed how odors impact barrel cortex activity in an active whisking context when they are presented with or without tactile gratings (Fig. 2a). Diverse response types were present in the barrel cortex population (Fig. 2b). In the presence of tactile inputs, odors could either

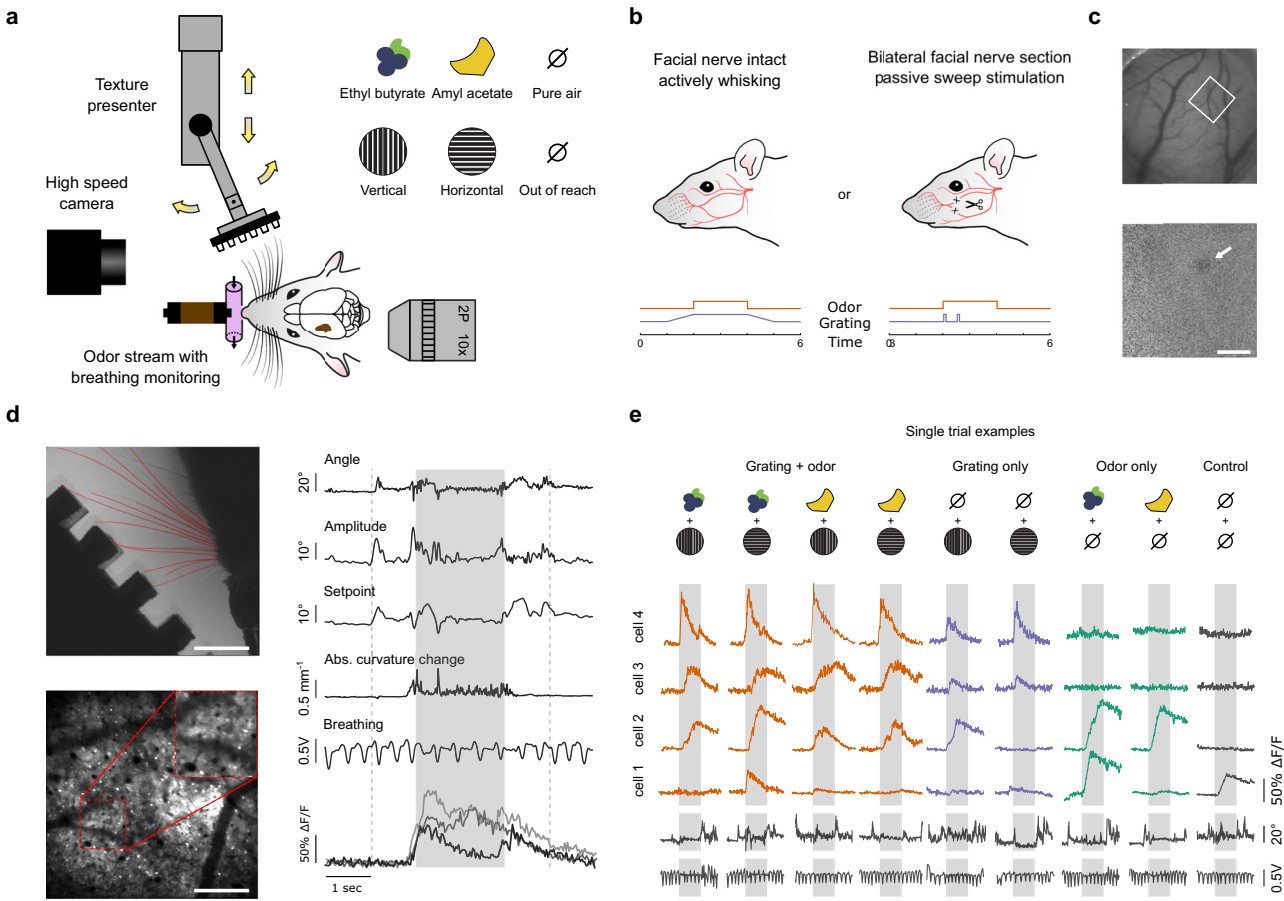

**Fig. 1 Calcium imaging in L2/3 of mouse barrel cortex during olfacto-tactile stimulation. a** Schematic of olfacto-tactile stimulation setup. A grating was moved in contact with the whiskers in either the vertical or horizontal orientation, while ethyl butyrate or amyl acetate was presented through a constant air stream. Whisking was recorded with high-speed videography (500 Hz) and breathing was monitored with a pressure sensor. **b** Schematics of the two recording conditions. Left: facial nerve intact, mice whisked freely on the grating brought to their whiskers together with stimulation timing. Right: facial nerve sectioned, the grating was swept on the whiskers while whisking was abolished. **c** Example of C2 barrel localization with intrinsic signal imaging used to confirm the location of GCaMP6s expression in barrel cortex (top: blood vessel image, bottom: intrinsic signal for the same field of view, scale bar: 1 mm, white square: two-photon field of view, arrow: C2 response). **d** Top left: example frame of whisker pad tracking with the vertical grating at its most extended position (scale bar: 7 mm). Bottom left: image of in vivo GCaMP6s expression in L2/3 (scale bar: 250 μm). Right: example of simultaneously recorded whisker pad kinematics, breathing, and raw $\Delta F/F_0$ traces. Dashed lines indicate onset and offset of grating approach; gray shading indicates the epoch of stimulation during which odors were present in the air stream. **e** Illustration of the nine combinations of tactile and olfactory stimuli with single-trial example $\Delta F/F_0$ traces from four neurons showing olfactory responses in the active context.

enhance or suppress tactile responses (Fig. 2b, first and second examples), and when presented alone odors could evoke excitatory or inhibitory responses (Fig. 2b, third and fourth examples). In some neurons, odors had the same impact whether a grating was presented or not (Fig. 2b, fourth example).

To quantify these effects at the population level, we compared the average population activity elicited by the gratings presented alone or paired with odors (Fig. 2c). For simplicity, these analyses were done by pooling trials with different odors together (for odor-specific analysis, see Fig. 6). We found no difference in population activity levels when odors were paired with gratings computed over the entire stimulation epoch. However, when we looked at single cells and compared the average activity to gratings alone and gratings with odors in the 2-s time window following stimulus onset, 21.7% of the stimulus-responsive neurons were detected as odor-modulated (Fig. 2d; significance threshold = 5%; Mann–Whitney test), with both enhanced and suppressed response types. Comparing the proportion of neurons impacted by odors with the proportion obtained from shuffling the trial labels confirmed that the proportion of neurons passing

the test cannot be explained by chance (Fig. 2e). To account for differences in activity levels between cells regardless of their tactile responsiveness, we computed a modulation index (Fig. 2f). Consistently, the distribution of indices revealed a significant impact of odors when compared with the distribution obtained from shuffled data (Fig. 2f), and an equal proportion of cells were enhanced or inhibited by odors (Fig. 2f, inset).

Similarly, odors presented alone did not impact average responses at the population level compared to double blank stimulation (no odor no grating) (Fig. 2g) but, at the single-cell level, significant odor responses could be detected in 18.34% (Fig. 2h–j) of the neurons, again with an equal proportion of enhanced and inhibited cells (Fig. 2j, inset). Note that due to the sound of the tactile presentation wheel, which was present in double blank trials, mice increase whisking at the beginning of double blank trials yielding a weak increase in neuronal activity in the barrel cortex (e.g., see Fig. 3). Together, these results show that in an active tactile context, odors modulate the activity of some barrel cortex neurons during both free whisking and whisking into tactile gratings.

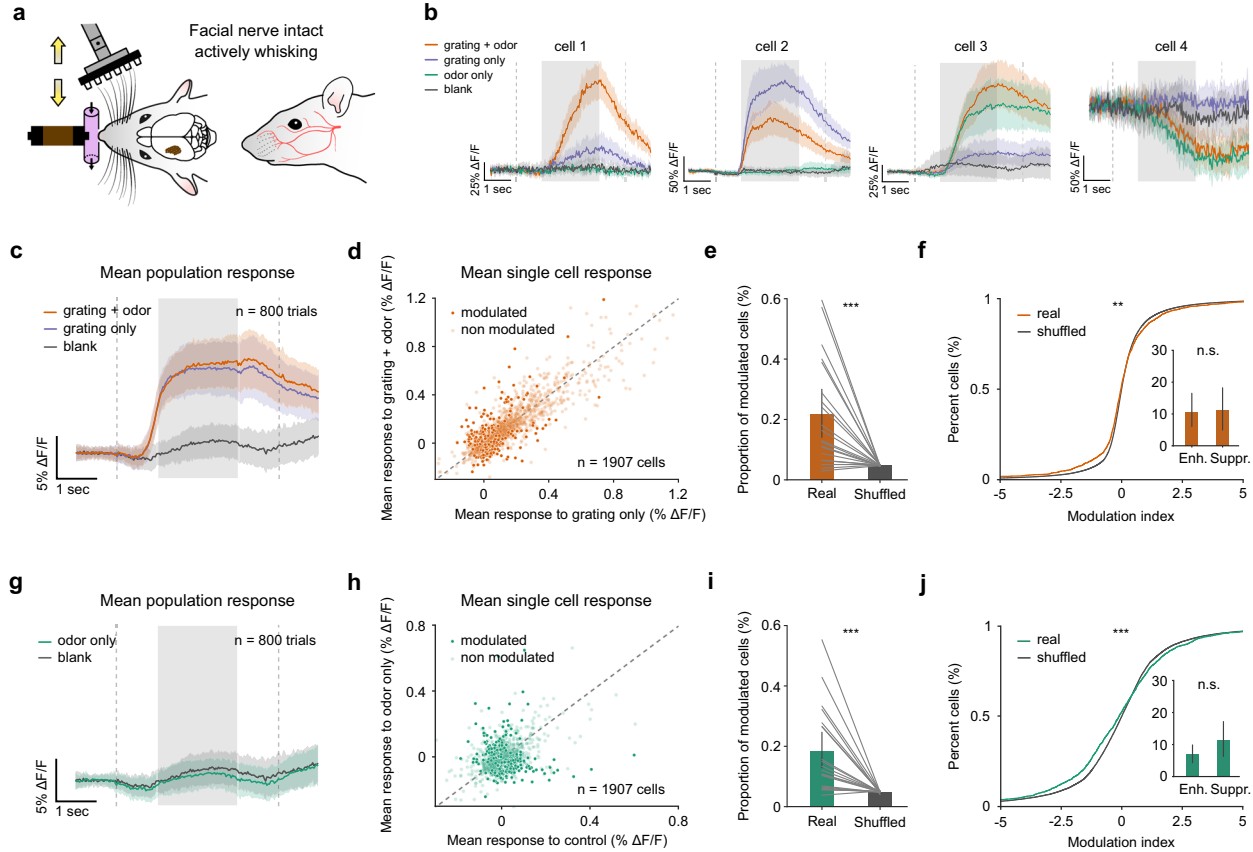

**Fig. 2 Odor-induced responses of L2/3 barrel cortex cells in the active stimulation context. a** Schematic of the stimulation condition with facial nerve intact. Gratings were brought in contact with the whiskers. **b** Average ΔF/F for four example neurons displaying diverse odor responses. **c–f** Responses to grating + odor stimulation compared with grating only. Stimuli were grouped into four categories: grating + odor (orange), grating only (blue), odor only (green), and control (gray). **c** Population averages to grating + odor trials (both orientations and both odors grouped together; orange) and grating-only trials (both grating orientations grouped together; blue); average to blank (gray) is shown for comparison. **d** Scatter plot of mean ΔF/F to grating+odor versus grating only for 1907 neurons. Neurons plotted in dark were significantly modulated ($P < 0.05$, Mann–Whitney $U$ test). Dashed line indicates diagonal. **e** Proportion of modulated neurons across sessions compared with shuffled conditions (21.71% (14.69, 29.58) for real data; 4.84% (4.80, 4.87) for shuffled data; $P = 2 \times 10^{-5}$; Wilcoxon signed-rank test, $n = 20$ sessions). **f** Cumulative distribution of modulation index for grating + odor versus grating alone ($P = 0.109$; Kolmogorov–Smirnov test, $n = 1907$ cells). Inset: proportion of enhanced and suppressed cells (mean enhanced = 10.35% (5.60, 15.77); mean suppressed = 11.36% (6.06, 17.60); $P = 0.793$; Wilcoxon signed-rank test, $n = 20$ sessions). **g–j** Response to olfactory stimulations alone compared with control. **g** Population averages to odor-only trials (green) and control (gray) (mean of odors only trials = 0.57% ΔF/F (−0.64, 2.01), mean of blank trials = 1% ΔF/F (0.28, 1.81) computed over stimulation epoch; $P = 0.394$; $n = 800$ trials from 20 sessions; Mann–Whitney $U$ test). **h** Same as (**d**) but for odor versus blank. **i** Same as (**e**) (18.34% (12.78, 24.76) for real data; 4.89% (4.84, 4.94) for shuffled data; $P = 1 \times 10^{-5}$; Wilcoxon signed-rank test, $n = 20$ sessions). **j** Same as (**f**) ($P = 2.3 \times 10^{-4}$; Kolmogorov–Smirnov test; inlet: mean enhanced = 6.97% (4.48, 9.98); mean suppressed = 11.36% (6.10, 17.52); $P = 0.398$; Wilcoxon signed-rank test, $n = 20$ sessions). Error bars and bands indicate 95% confidence intervals for the mean throughout the figure. All tests are two-sided.

**Odors impact whisker dynamics**. The mechanisms by which odors can potentially impact barrel cortex activity are numerous. One possibility is that odors motivate whisker movements which generate tactile inputs to the barrel cortex as reafferent sensory signals. To explore this possibility, we tracked whisker movements using high-speed videography and an automated whisker tracking algorithm[20], which allowed us to robustly detect and trace ~12 individual whiskers within the full pad at each time point. From this tracking, we calculated the average amplitude, setpoint, and absolute curvature change across time (Fig. 3a). During trials without tactile gratings, odors caused a slight but significant increase in average amplitude, setpoint, and absolute curvature change both within sessions (Fig. 3b–d) and between sessions (Fig. 3b–d). Similarly, in the presence of tactile gratings, odors caused a significant increase in average amplitude and absolute curvature change (Fig. 3b–d). In this condition, setpoint

did not change significantly in the presence of odors despite a tendency towards an increase. This is likely due to the restriction of the full whisking range by the gratings. This analysis shows that odors lead to significant changes in whisking parameters. As there is an intricate link between whisking behavior and sniffing, we also checked if breathing was impacted by odors. Breathing amplitude (Fig. 3e), measured in a 2-s time window starting at odor onset, was significantly reduced in the presence of odors compared to the conditions without odors. A similar tendency could be found in breathing frequency (Fig. 3f), but the reduction was not robust enough to be significant. Together, these data show that mice modify their oro-facial motor programs when presented with odors. Because whisking impacts barrel cortex activity[21,22], this implies that at least part of the odor-related activity in the barrel cortex could be due to modulations of whisking.

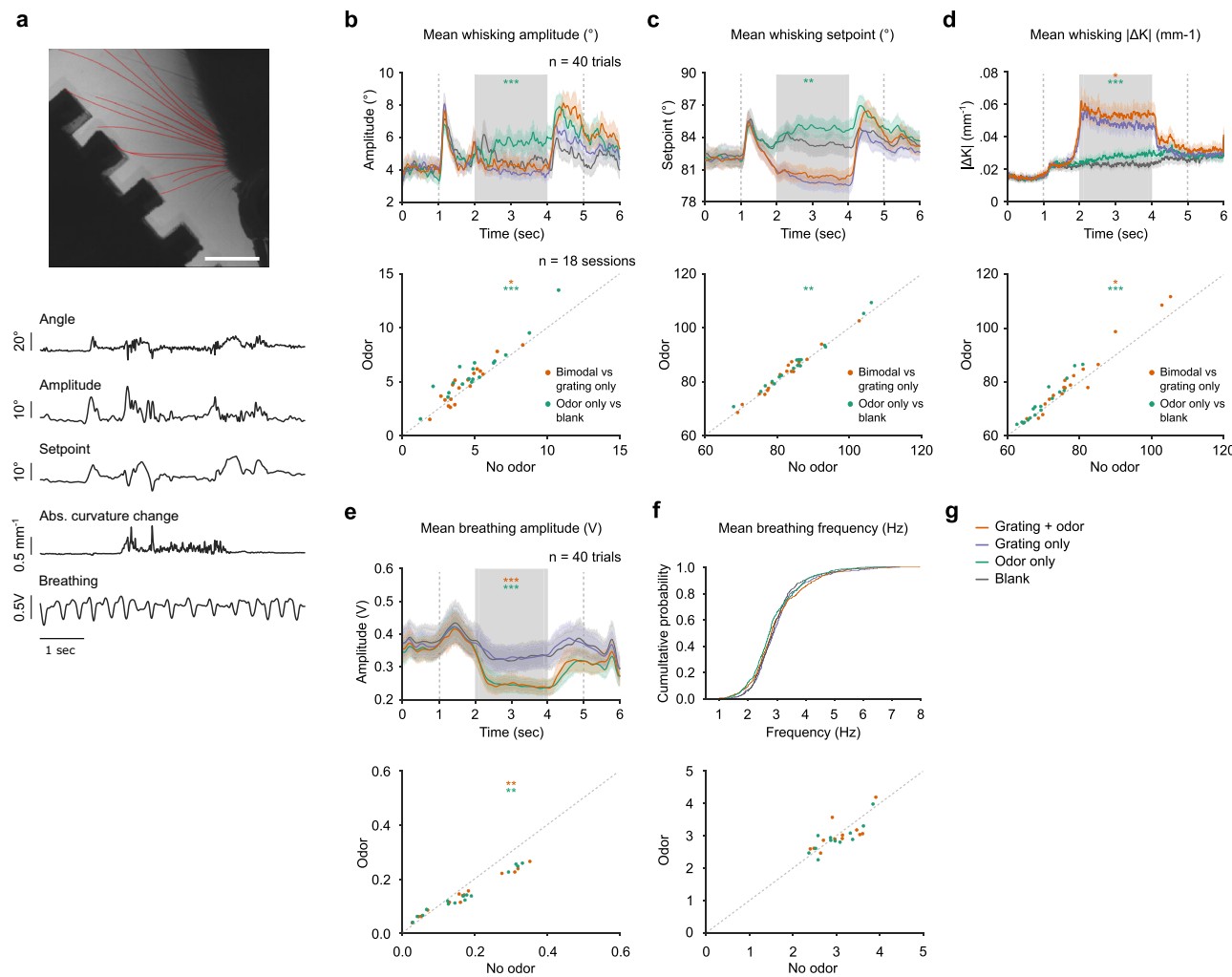

**Fig. 3 Olfactory stimulations decrease breathing and increase whisking kinematics. a** Example frame of whisker tracking with traces of derived whisking kinematic parameters and breathing shown below (scale bar: 7 mm). **b** Top: mean whisking amplitude across time within sessions. Bottom: scatter plot of mean whisking amplitude between sessions at stimulus presentation. Whisking was analyzed over the 2-s stimulation epoch indicated by gray shading (orange: bimodal versus grating only; green: odor alone versus blank). **c** Top: mean setpoint within sessions. Bottom: mean setpoint between sessions. **d** Mean absolute curvature change within sessions. Bottom: mean absolute curvature change between sessions. **e** Mean breathing amplitude within sessions. Bottom: mean breathing amplitude between sessions. **f** Cumulative distribution of breathing frequency within sessions. Bottom: mean breathing frequency between sessions. *n* = 18 sessions from 10 mice; *P < 0.05, **P < 0.01, ***P < 0.001; Wilcoxon signed-rank test; see Supplementary Tables 1 and 2 for detailed statistics. **g** Legend to (**b**–**f**) top. Error bands indicate 95% confidence intervals for the mean throughout the figure. All tests are two-sided.

**Odor-induced responses persist in the absence of peripheral interaction**. To examine if the impact of odors on barrel activity depends on mechanisms that go beyond the peripheral modulation of whisking by odors, we performed recordings in a passive stimulation context in which whisking was prevented by bilateral sectioning of the inferior and superior buccal branches of the facial nerve (Supplementary Fig. 3). In this case, because the mice were not able to explore the tactile stimuli, the gratings were swept once forward and backward through the whisker pad (Fig. 4a). This experimental approach generated robust tactile responses in S1 while removing any potential for odor-induced whisking that results in reafferent sensory signals in the barrel cortex. Similar to the active context (Fig. 2b), odors impacted barrel cortex activity with both enhancement and suppression (Fig. 4b). At the population level, odors still had no effect on the trial-averaged total activity, except for a small increase that was significant within sessions (Fig. 4c, g), but not robustly observed between sessions (Fig. 4c, g). At the level of single cells, again, we observed significant modulation of responses by odors with equal

proportions of enhanced and suppressed cells. For each cell, we compared the average activity to the grating sweeps alone versus the sweeps with odors. In the absence of whisking, average ΔF/F's were still detected as significantly modulated by odors in 8.99% of the stimulus-responsive neurons (Fig. 4d, e; significance threshold = 5%; Mann–Whitney *U* test), a proportion that is still significantly above chance level (Fig. 4e) and with a significant bias towards enhancement (Fig. 4f, inset). Consistently, the distribution of modulation indices showed a significant impact of odors when compared to shuffled data (Fig. 4f). Interestingly, the fraction of odor-modulated neurons in the absence of whisking was significantly lower than in freely whisking mice (8,99% against 21.7%, *P* = 0.012, Mann–Whitney *U* test), indicating that the peripheral modulation of whisking by odors generates a significant fraction of odor-related activity in barrel cortex but not all of it.

In the same way, odors presented alone in the passive context also modulated barrel cortex activity with balanced enhancement and suppression. The average population response was equivalent

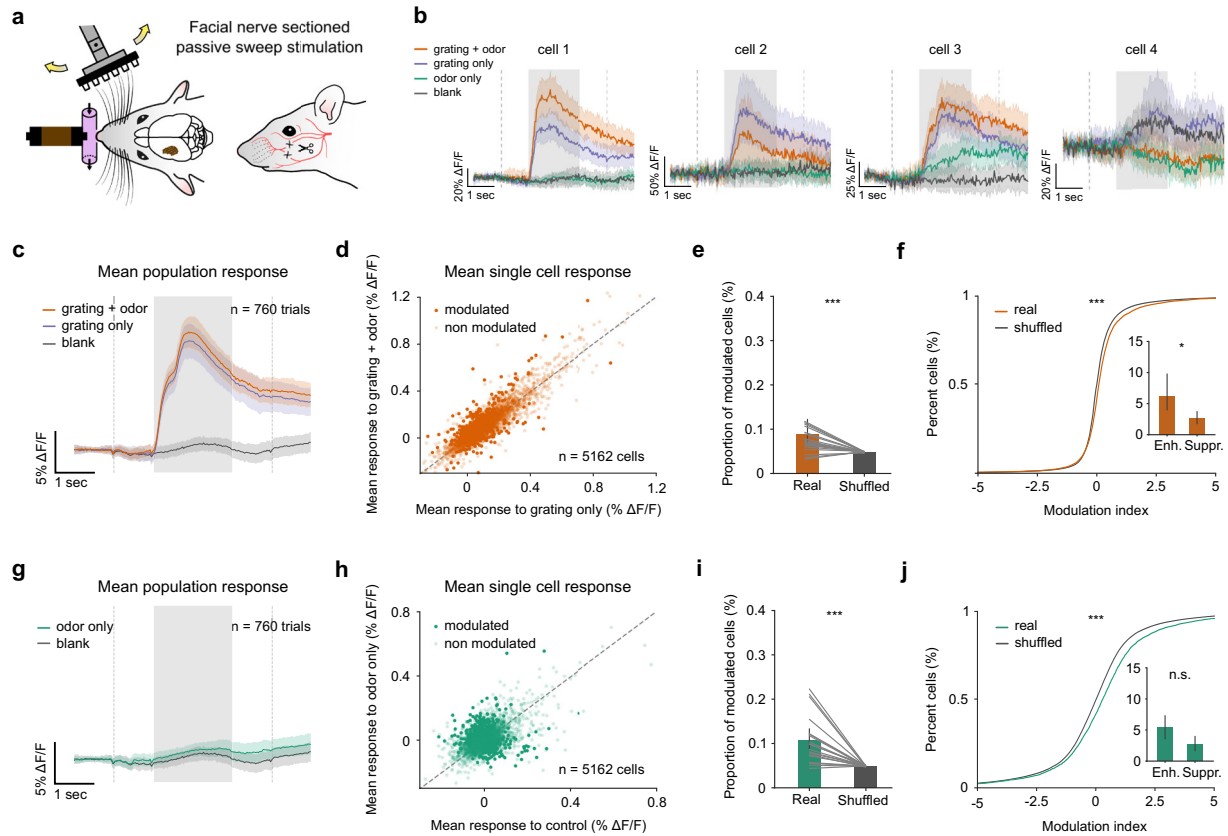

**Fig. 4 Odor-induced activity in the barrel cortex persists in the absence of whisking. a** Schematic of the stimulation context with facial nerve sectioning. Gratings were brushed against the whiskers in a back and forth sweep. **b–j** Same as in Fig. 1b–j. **b** Average ΔF/F for four example neurons displaying diverse odor responses. **c** Population averages to grating + odor trials (orange) and grating-only trials (blue); average to control (gray) is shown for comparison (mean of bimodal trials = 11.22% ΔF/F (10.78, 11.66); mean of gratings only trials = 10.32% ΔF/F (9.81,10.73); $P = 0.001$ within sessions; $n = 760$ trials; Mann–Whitney $U$ test; $P = 0.28$ between sessions; $n = 19$ sessions; Wilcoxon signed-rank test). **d** Scatter plot of mean ΔF/F to grating+odor versus grating only for 5162 neurons. Neurons plotted in the dark are significantly modulated ($P < 0.05$, Mann–Whitney $U$ test). Dashed line indicates diagonal. **e** Proportion of modulated neurons across sessions compared with shuffled conditions (8.99% (7.79, 10.57) for real data; 4.81% (4.75, 4.87) for shuffled data; $P = 0.0006$; Wilcoxon signed-rank test, $n = 19$ sessions). **f** Cumulative distribution of modulation index for grating + odor versus grating alone ($P = 1.15 \times 10^{-17}$, Kolmogorov–Smirnov test, $n = 5162$ cells). Inset: proportion of enhanced and suppressed cells (mean enhanced = 6.16% (3.89, 9.70); mean inhibited = 2.83% (1.68, 3.91); $P = 0.047$; $n = 19$ sessions; Wilcoxon signed-rank test). **g** Population averages to odor-only trials (green) and control (gray) (mean of bimodal trials = 0.91% ΔF/F (0.62, 1.2); mean of gratings only trials = 0.29% ΔF/F (0, 0.61); $P = 0.001$ within sessions; $n = 760$ trials; Mann–Whitney $U$ test; $P = 0.25$ between sessions; $n = 19$ sessions; Wilcoxon signed-rank test). **h** Same as (**d**). **i** Same as (**e**) (10.81% (8.55, 13.27) for real data; 4.89% (4.84, 4.94) for shuffled data; $P = 0.0001$; Wilcoxon signed-rank test, $n = 19$ sessions). **j** Same as (**f**) ($P = 5.91 \times 10^{-120}$; Kolmogorov–Smirnov test, $n = 5162$ cells; inlet: mean enhanced = 7.24% (4.94, 9.84); mean suppressed = 3.57% (2.08, 5.26); $P = 398$; $n = 19$ sessions; Wilcoxon signed-rank test). Error bars and bands indicate 95% confidence intervals for the mean throughout the figure. All tests are two-sided.

for odor alone and double blank trials (Fig. 4g) with a small enhancement within sessions (Fig. 4g). A proportion of 10.81% neurons were significantly modulated by odors in this passive context (Fig. 4h, j; significance threshold = 5%; Mann–Whitney $U$ test), which was also significantly above the chance level (Fig. 4i). Modulation indices also showed a significant impact of odors presented alone on the population of recorded cells (Fig. 4j). Again, the absence of whisking produced a decrease in the fraction of cells responsive in the odor-alone context, although not significantly due to variability across sessions (10.81% against 18.34%, $P = 0.118$, Mann–Whitney $U$ test). Together, this observation shows that along with modulation of whisking behavior, there is a second mechanism by which odors impact barrel cortex activity that is still active when whisking is abolished.

**Odor-related activity in S1 does not come through cholinergic inputs.** Since whisking is not the sole factor explaining the odor-related activity in S1, we examined if it could result from

cholinergic inputs to the cortex. Cholinergic inputs to the cortex from the basal forebrain have very specific effects on excitatory and inhibitory neurons[23–26] and are known to play a role in attentional modulation based on the behavioral context[27–30]. To assess whether they play a role in odor-related modulation of barrel cortex activity, we performed dual local injections of a nicotinic and a muscarinic receptor antagonist (1 mM injection of atropine and mecamylamine, Fig. 5a) in the barrel cortex shortly before running our olfactory-tactile stimulation protocol in animals with their facial nerve sectioned. This pharmacological perturbation of cholinergic signaling did not have a significant effect on the number of cells modulated by odors presented with or without tactile stimulation (Fig. 5b). Thus, we found no evidence that cholinergic signaling underlies odor-driven barrel cortex activity.

**Odor-specific olfactory information in the barrel cortex weakly impacts tactile representations.** While the significant proportion of neurons that can be detected as odor-modulated suggests the

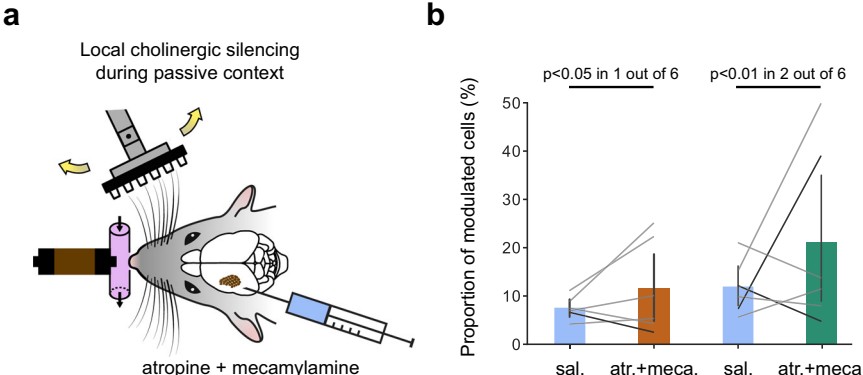

**Fig. 5 Pharmacological blockade of cholinergic transmission in the barrel cortex does not reduce the proportion of odor-modulated cells. a** Schematics of local silencing of cholinergic transmission with 1 mM injections of atropine (atr.) and mecamylamine (meca.) in barrel cortex during the passive stimulation context. **b** Proportion of modulated cells with and without unilateral cholinergic silencing (mean saline = 7.57% (5.93, 9.34); mean atr.+meca. = 11.59% (4.87, 18.63); $P = 0.463$; grating + odor versus grating only (orange); mean saline = 11.98% (8.17, 16.28); mean atr.+meca. = 21.24% (8.93, 35.75); $P = 0.6$; odor only versus blank (green); $n = 12$ sessions from three mice; Wilcoxon signed-rank test). Black lines indicate pairs of sessions with a significant change in the proportion of modulated cells (Chi-square test). Error bars indicate 95% confidence intervals for the mean. All tests are two-sided.

presence of odor information in barrel cortex, the intrinsic uncertainty about whether or not a given odor modulation is a true or false detection makes the quantification of the actual level of information difficult. We therefore evaluated the information carried by odor-related activity in the barrel cortex in the passive context when whisking is abolished using centroid classifiers with a stratified 20-fold cross-validation, trained on population activity vectors to discriminate various stimulation conditions. This approach is independent of statistical thresholds as it considers all neurons whether or not their modulation by the odors is statistically detectable. We also used the same method to quantify to what extent this information interacts with tactile representations. We first asked whether the presence of an odor was discernible in the population activity by training a classifier to discriminate bimodal trials from tactile-only trials or to discriminate odor-only trials against blank trials in mice with facial nerve sectioning (Fig. 6a, b; same analysis for the active whisking context shown in Supplementary Fig. 4a–d). A proportion of 86.4% of the odor only against blank trials and 71.2% of the bimodal against tactile-only trials were correctly classified using activity from 1 to 2 s after stimulus presentation. This high performance of the classifier in both cases (yet smaller than for a tactile discrimination, Fig. 6c) shows that information about the presence or absence of an odor is robustly encoded in the barrel cortex. Next, we examined if the olfactory information present in the barrel cortex was specific to the type of odor presented. Amyl acetate and ethyl butyrate are two very distinct chemicals which are well-discriminated in circuits of the olfactory system[31]. Interestingly, when training classifiers to discriminate amyl acetate and ethyl butyrate trials, classification performance was well above chance level with 68.5% of trials correctly classified (Fig. 6d). These effects were robust enough to be present across sessions (Supplementary Fig. 4e–l). Olfactory information was also present during silencing of cholinergic inputs (Supplementary Fig. 5) and was not due to a systematic bias in mean population activity (Supplementary Fig. 6). Hence, odor-related activity in the barrel cortex is sufficiently precise to decode some information about odor identity.

This observation raises the question whether the presence of olfactory information in the barrel cortex has any impact on the tactile representations or if the two representations are independent. To answer this question, we quantified whether classifiers discriminating the two different tactile grating orientations

presented to the animal were impacted by odor-driven activity. Tactile grating orientation classification could be easily performed by barrel cortex populations, especially in facial nerve-sectioned animals (Fig. 6c). To evaluate the impact of coincident odors, we trained a classifier to discriminate population activity for the two grating orientations using a training set of trials without odors. We then measured the performance of the classifier using a test set of trials with odors and a test set of trials without odors. When using a global neuronal population merging all recording sessions for this analysis (as in Fig. 6c), the classifier score for discrimination was 100% in both cases, suggesting no strong impact of odors on tactile coding. For a more sensitive measurement, we also performed this analysis on single sessions. In this case again, we found no systematic difference between the classification scores obtained in the presence and the absence of odor (Fig. 6e), confirming that odor-evoked activity does not strongly modify tactile representations along dimensions that are important for their discrimination (Fig. 6e). Conversely, classifiers trained on odor-only trials to discriminate odor identity were not robust to the presence of tactile stimuli (Fig. 6f). The same result was observed in the active context (Supplementary Fig. 7). This indicates that, for the much weaker odor representations, the neural activity dimensions that are relevant to extract odors also represent tactile information as seen in Fig. 4d. Therefore, despite the lack of impact of odors on the global population of barrel cortex neurons (Fig. 6e), there exists a subspace of barrel cortex activity in which odor information is present and within which tactile representations interact with olfactory representations.

**Olfactory information in barrel cortex is not explained by facial movement behavior.** To investigate whether the odor-related subspace of barrel cortex activity is specific to olfactory information or broadly relates to the state of our awake animals which can be itself modified by odors, we further evaluated the specificity of odor information present in the barrel cortex. First, we increased the complexity of the odor space by extending the set of odors to be presented. We performed a second set of 11 barrel cortex imaging sessions in which we randomly presented five different monomolecular odors (amyl acetate AA, ethyl butyrate EB, heptanal, limonene, hexanone) and four different mixtures (80/20; 40/60; 60/40; 20/80%) of AA and EB. Whiskers

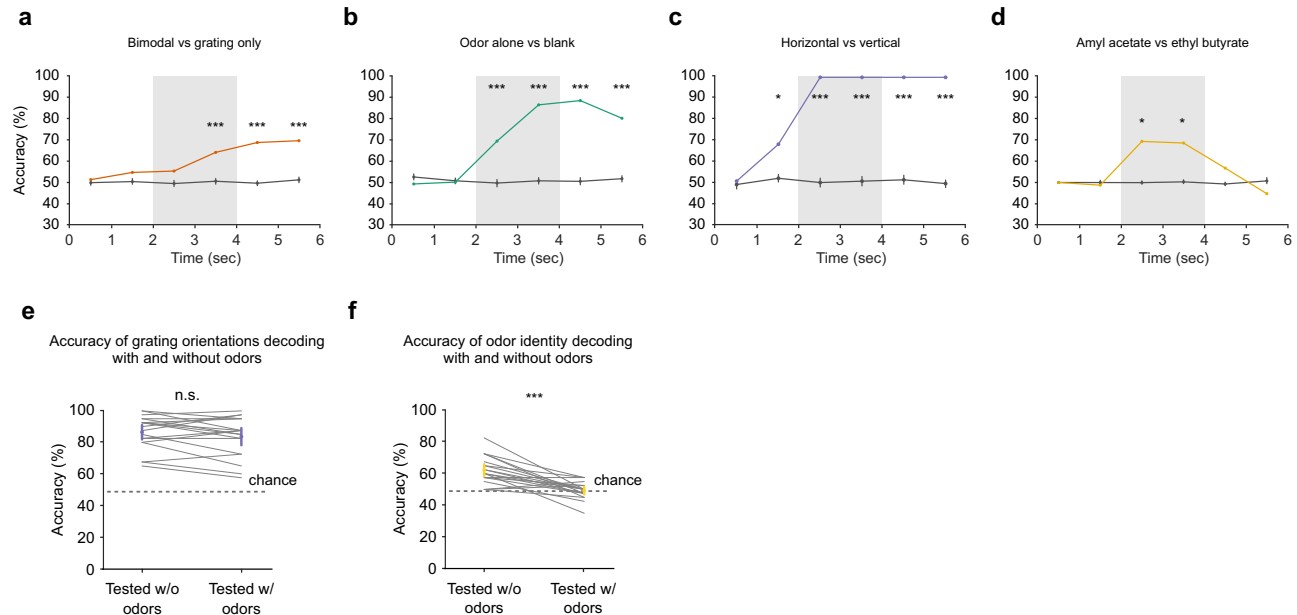

**Fig. 6 Tactile and olfactory codes are independent in wS1. a** Accuracy of centroid classifiers decoding bimodal from touch-only stimuli based on S1 activity in the passive context averaged over 1-s time bins. Performance for shuffled labels is shown in gray. Error bars indicate 95% confidence intervals for the mean. $P$ values were obtained as the location of the mean accuracy in a distribution of 1000 shuffles (two-sided, no correction for multiple testing); $*P < 0.05$, $**P < 0.01$, $***P < 0.001$. Shading indicates stimulus presentation. **b**–**d** Same as in **a** for decoding of odor only and blank stimuli, grating orientation, and odor identity, respectively. **e** Accuracy of centroid classifiers trained to decode grating orientations without odors and showing no impairment when tested in the presence of odors (mean without odors = 88.5% (83.4, 93.1); mean with odors = 86.3% (81.0, 91.0); $P = 0.401$; Wilcoxon signed-rank test). **f** Same as in **e** for decoding odor identity with classifiers trained without touch and going to chance level when tested with touch (mean without touch 63.9 = % (60.6, 67.1); mean with touch = 49.2% (46.4, 51.9); $P = 2.62 \times 10^{-8}$; Wilcoxon signed-rank test). All tests are two-sided.

were again immobilized by nerve sectioning (Fig. 7a). We used an acousto-optic microscope allowing the recording of four different planes (478 × 478 μm each) interleaved by 60 μm (Fig. 7a) and within which 1190 neurons could be isolated on average in each session. Due to the absence of tactile stimulation in this experiment, out of 13168 recorded neurons only 6.7% were stimulus-responsive ($P < 0.05$; Kruskal–Wallis test), a proportion consistent with and larger than the small fraction of odor-responsive neurons found within the stimulus-responsive population in previous experiments (Fig. 4). However, sparse but clear examples of cells displaying odor selectivity could be isolated as shown in Fig. 7b. In line with these observations, population representations of odors in the barrel cortex allowed decoding odor identity significantly above chance (36% accuracy; chance = 20%) as estimated with a cross-validated support-vector machine classifier applied to single-trial responses (Fig. 7c). For a single session, decoding performance was low, owing to the high dilution of odor responses in the barrel cortex, but still above the decoding accuracy obtained after shuffling of odor identity, although not significantly (Fig. 7c). This information also captured some similar relationships between odors. Indeed, if we asked a classifier to categorize responses to intermediate mixtures of AA and EB, the classifier consistently assigned mixtures to the odor corresponding to the dominant mixture component (Fig. 7d). Thus, specific information about odor identity is provided by odor-evoked activity in the barrel cortex, although it is diluted and partially covered by neuronal response variability.

To investigate if this odor-specific information relates to behavior, brain states, or only to odors, we evaluated if behaviors observed during odor presentation could explain odor responses in the barrel cortex. We focused on facial movement behaviors as they have been reported to tightly relate to rapid fluctuations of cortical activity states[32] and analyzed videos of the mouse's face

made during the odor stimulation sessions (Fig. 7a). To condense this rich information, we performed a principal component analysis on the videos based on the Facemap algorithm[32]. Individual principal components captured global motion as well as motion of different parts of the face including the eye and jaw whose motility was preserved by our resection of the buccal and marginal mandibular branches of the facial nerve that drives whisking (Supplementary Fig. 8 and Fig. 1). Similarly, population activity across the absence and presence of odor could also be decomposed into principal components (PCs). As previously observed[32], the first PC of neuronal activity does not dominate over other components and a large number of PCs are necessary to account for the full variance of cortical activity, attesting to the high-dimensionality of barrel cortex activity (Fig. 7e). Visual inspection of the first PCs for facial behavior (Fig. 7f and see Supplementary Fig. 8 for PC weights) indicated that movement was only partially correlated to barrel cortex activity, in particular during odor presentation. In line with this, plotting trial-averaged PCs of facial behavior against neural response PCs during odor presentation did not reveal a strong resemblance between the time courses of facial behaviors and the time course of neuronal activity (Fig. 7g), suggesting that there is little information about odors shared between population responses and facial movement behavior. To directly test this, we measured the performance of classifiers based on the first 500 PCs of facial movement behavior. The performance of the classifier was at chance levels whether we were considering single animals or facial features concatenated across animals (Fig. 7h), indicating that facial behavior does not contain information about odor identity. Including fewer PCs than 500 did not improve the decoding and a similar level of accuracy was obtained using respiration or whisking parameters (Supplementary Fig. 9). Therefore, odor information in the barrel cortex cannot be explained by odor-dependent changes in facial behaviors.

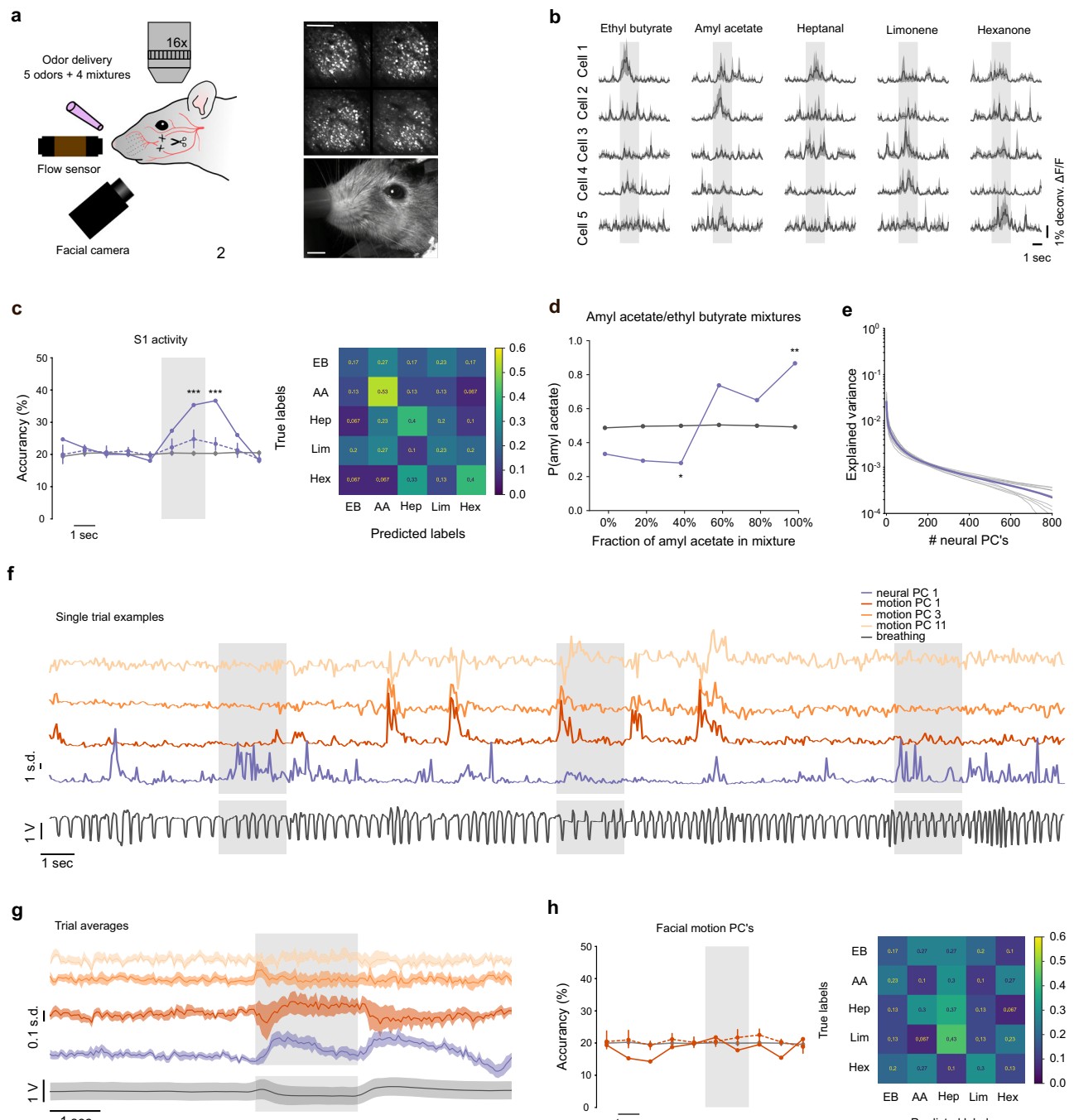

**Fig. 7 Olfactory information in the barrel cortex is not explained by facial behavior. a** Schematic of the multi-odor protocol (left) and max projection of a sample field of view for two-photon imaging of four simultaneous planes Scale bar 200 μm. (right). Facial behavior was monitored with a video camera at 20 Hz. Scale bar 2 mm. **b** Mean deconvolved calcium signal for five odors and five example barrel cortex neurons. Shading indicates odor presentation. Error bars indicate 95% confidence intervals for the mean traces. **c** Left: classifier accuracy for the 5 odor identities. Purple line: classification with the full dataset (13,168 neurons). Dashed line: mean accuracy of single session classifications. Gray line: chance level estimated on shuffled data. Right: confusion matrix for the full dataset classifier at the second time bin of stimulus presentation. EB ethyl butyrate, AA amyl acetate, Hep heptanal, Lim limonene, Hex hexanone. **d** Accuracy of a classifier discriminating amyl acetate from ethyl butyrate based on barrel cortex activity (full dataset) and categorization scores of the same classifier for four mixtures of the two odors. The classifier was not trained on mixtures. **c**, **d** \*\*$P < 0.01$, \*\*\*$P < 0.001$ (two-sided $P$ values, based on the distribution of 1000 shuffles. Minimum $P$ value: 0.001, no correction for multiple testing. **e** Spectrum of the fraction of explained variance for the principal components (PCs) of neuronal activity (mean across 11 sessions). **f** Loadings of three facial behavior PCs and of the first neuronal activity PCs for a short time sample in an example recording session. **g** Average loadings for the same PCs as in **f**, across all odor presentations of the recording session. **h** Left: decoding accuracy of odor identity using the first 500 facial behavior PCs. Chance level performance is obtained whether the classification is done in single sessions (dashed line, $n = 12$ sessions) or after pooling PCs of all sessions (plain line). Right: confusion matrix of the pooled-session classifier. Error bars and bands indicate 95% confidence intervals for the mean throughout the figure. All tests are two-sided.

## Discussion

Using two-photon calcium imaging during olfactory-tactile stimulation in head-fixed mice with an intact whisker pad, we observed that the barrel cortex encodes information about odor identity in parallel with the tactile representations of the proximal environment. We identify two mechanisms for odor-induced modulations of barrel cortex activity: a peripheral one, related to altered whisking when an odor is present (Figs. 2 and 3), and another, presumably of central origin, that is independent of whisking and cholinergic signaling (Figs. 4 and 5). Together, these two mechanisms affect about one-fifth of stimulus-responsive barrel cortex neurons in freely whisking animals, while the latter, central mechanism impacts only about one-tenth of the stimulus-responsive neurons. Despite this modest proportion, whisking-independent odor representations are sufficient to reliably decode not only the presence of an odor, but also its identity, making it unlikely that these modulations reflect intact efference copies of whisking programs after facial nerve sectioning (Fig. 6). In turn, it is unclear whether whisking-dependent effects themselves inform about odor identity as they could not be disentangled from the whisking-independent effects. However, odor decoding was not significantly improved when mice could whisk (Supplementary Fig. 4). Moreover, the relevance of whisking-dependent effects for behavior is difficult to interpret. In our experimental conditions, they reflect global changes in the dynamics of the whisker pad which most probably corresponds to an increase in the engagement of the animal to explore its near environment whether it contacts an object or not (Fig. 3). Although these reafferent signals impact the barrel cortex, it is possible that they are compensated by efference copies from motor centers[33–35]. In a goal-directed context, these effects may also be canceled by tighter control of whisking during behavior[36,37].

Whisking-independent effects may reflect in contrast a more central cross-modal signaling mechanism. Our pharmacological manipulations (Fig. 5 and Supplementary Fig. 5) suggest that they do not depend on cholinergic signaling, indicating that they do not reflect attentional modulation mediated by this pathway. This does not rule out signaling by other neuromodulatory pathways which could play a role as suggested by the fact that both whisking and breathing are affected by the presence of an odor in our experimental conditions[38–41]. For these effects to be entirely driven by neuromodulatory signaling, it would mean that neuromodulatory inputs to the barrel cortex are odor-identity-specific, because amyl acetate and ethyl butyrate trials (Fig. 6 and Supplementary Fig. 4) and five different odors can be discriminated based on barrel cortex activity (Fig. 7). In fact, recent reports suggest that the internal states which modulate cortical activity are largely reflected in facial behaviors[32]. Because we could not retrieve odor information from facial behaviors, it is unlikely that neuromodulation states are enough to explain all odor responses in the barrel cortex.

Outside of neuromodulatory sources, there are several other pathways that could introduce odor information into barrel cortex representation. Although there are no direct inputs to the barrel cortex from the piriform cortex[12,42], they are bidirectionally connected to common associative areas, in particular the perirhinal cortex[12,43–46]. The perirhinal cortex, which is located just above the piriform cortex on the ventrodorsal axis, is an associative area that has been implicated in multimodal object recognition[47–50]. To the best of our knowledge, no study has addressed its potential role in olfactory-tactile integration, but in the rodent, it could be a multimodal hub that combines these two sensory modalities among others. In this case, the odor-evoked activity we observe in the barrel cortex would represent feedback information from an associative area. Alternatively, barrel cortex receives inputs from several thalamic nuclei, including from secondary thalamic regions which themselves are known to receive multimodal inputs[51].

Independent of the pathway mediating it, the functional significance of olfactory signaling in the barrel cortex remains an open question. We observed that olfactory activity in the barrel cortex contains an equal amount of inhibitory and excitatory responses and builds neural representations which have relatively little impact on global tactile information (Fig. 6e). Coincident olfacto-tactile stimulations neither sharpen tactile representations in the barrel cortex as has been observed recently in the visual cortex with sounds[14,15], nor do they improve detection of multimodal coincidence. The advantage of this coding scheme is that the barrel cortex can access olfactory information without perturbations of the tactile code. However, we observed that the weaker olfactory representations of the barrel cortex interact with tactile representations (Fig. 6f). This suggests that while the dominant coding space for tactile information is unaffected by odors, there exists a small subspace of barrel cortex activity that encodes the two modalities in an intricate manner. This is consistent with the recent observation that a primary sensory cortex encodes its main sensory modality and behavioral or contextual information in largely distinct subspaces of population activity[32]. Our results indicate that within this contextual information, there is a dedicated channel related to cross-modal (here olfactory) information which does not fully overlap with behavioral information, as in our case behavior did not reflect specific odor information (Fig. 7h). The existence of an olfactory-tactile subspace in the barrel cortex could enable the emergence of cross-modal associations in specific behavioral contexts which require information from both modalities[52]. The fact that odors have both inhibitory and excitatory effects (Figs. 2 and 4) suggests that odors could suppress some aspects of the tactile representations and boost others. Together our results demonstrate a site of convergence for olfactory and tactile information early in the sensory processing hierarchy and open interesting avenues to study the role of brain-wide interactions in processing two sensory modalities that are crucial in rodents' daily life.

## Methods

**Cranial window implantation and viral injections**. All procedures were conducted in accordance with protocols approved by the French Ethical Committees #59 and #89 (authorizations APAFIS#9714-2018011108392486 v2 and APAFIS#27040-2020090316536717 v1). We used 8 to 12-week-old C57BL/6J male and female mice housed 1–4 per cage, in a normal light/dark cycle (12 h/12 h). Cranial window implantation and viral injections were performed under isoflurane anesthesia (1.3–1.7%) with body temperature maintained constant at 37 °C using a regulated thermal blanket with a rectal probe (Rodent Warmer X1, Stoelting). A craniotomy of 4 mm in diameter was drilled over the barrel cortex on the left hemisphere. Four injections of 200 nl of AAV1-syn-GCaMP6s ($1 \times 10^{-12}$ vg ml$^{-1}$), obtained from Vector Core (Philadelphia, PA, USA), were performed with glass micropipettes and a programmable oil-based injector (Nanoliter 2000 & Micro 4; World Precision Instruments) at 30 nl.min$^{-1}$ around the C2 barrel column at AP—1.6, ML—3.3, DV 0.5[19]. The craniotomy was sealed with a glass window comprising two circular coverslips bound together with optical glue (5 and 3 mm diameter) and a metal post for head-fixation was implanted using dental cement (Super-Bond C&B, Sun Medical Co. Ltd.).

**Facial nerve transection**. For experiments in the passive condition, whisking was prevented by bilateral sectioning of the buccal and marginal mandibular branches of the facial nerve[53,54]. Mice were anesthetized under isoflurane with their temperature monitored similarly to the cranial window implantation. A 3 mm cut of the skin was performed to expose the two branches of the facial nerve which were sectioned with microsurgical scissors. The cut was then closed with a nylon suture.

**Intrinsic optical imaging**. The location of the GCaMP6s expression locus in the barrel cortex was validated with intrinsic optical imaging under isoflurane anesthesia (1%; SomnoSuite, Kent Scientific) on a thermal blanket. The signal was obtained under 625 nm LED illumination and images of the vasculature over the same field of view were taken under 480 nm LED illumination. Reflected light was acquired with a CCD camera (GC651MP, Smartek Vision) equipped with a 50 mm objective (Fujinon, HF50HA-1B, Fujifilm) on a 656 × 496 pixel region and

resolution of 5.58 μm per pixel at 15 fps through a cranial window 1–2 weeks after implantation. Four seconds after imaging onset, the C2 whisker was deflected by a piezoelectric bender (PI PICMA Bender) at 10 Hz for 4 s following a sinusoidal wave along the rostrocaudal axis, for twenty trials with 8 s inter-trial intervals. Change in reflectance was computed as $(R_{stim} - R_{base})/R_{base}$ where $R_{stim}$ and $R_{base}$ are averaged over the 4 s of stimulus presentation and baseline, respectively. Response images were averaged across all deflections. Mice whose intrinsic response did not coincide with the GCaMP6s expression locus were excluded from further analysis.

**Odor delivery and grating presentation.** Odorants were from Sigma-Aldrich and delivered with a custom-made olfactometer (Supplementary Fig. 1a). Mass flow controllers were used to pass airflow through small bottles (Wilmad ML-1490-702, SP Scienceware) filled with 20 ml of odor solution diluted at 0.1% in mineral oil and 1% for the multi-odor experiment (Fig. 7). The total flow was constant ($1 \, l \, min^{-1}$) and the snout of the animal was placed inside a confined air stream to prevent movement of the whiskers. Photoionization detector measurements were made to assess the temporal precision of delivery; to obtain a stable concentration during stimulus application, we ensured that the flow was stationary with a 5-s bubbling period before the stimulus was presented (Supplementary Fig. 1b, c). Binary odor mixtures of amyl acetate and ethyl butyrate (80/20, 60/40, 40/60, 20/80%) were produced by differential air-flow dilution using the pair of delivery mass flow controllers (Supplementary Fig. 1a).

Oriented gratings were made of five ridges of 3.5-mm thickness and spacing on a 3d printed PLA disk of 35 mm diameter. Gratings were actuated with a custom-made presentation wheel consisting of two stepper motors (42BYG, Makeblock) mounted on a linear stage (eTrack, Newmark) and controlled with an Arduino. In the active condition, the gratings were brought from their starting location to the tips of the whiskers 4 cm away, reaching their most extended position after 1 s. In the passive condition, the gratings were brought by the linear stage at the same speed, rotated 10° anterior to the mouse. Once the linear stage reached its final position, the gratings were rotated 30° forward at a speed of 35 cm/s (thus sweeping over the whiskers in 100 ms), then rotated 30° backward 500 ms after initiation of the first sweep.

In the active condition, the odor delivery onset at the snout location (measured with a PID; 44 ms latency after shuffle valve switch) was synchronized with the moment where the gratings reached their most extended position. As mice in the active condition tend to whisk to fetch the approaching grating cued by the sound of the linear stage, whiskers could make first contact with the gratings up to 400 ms before odor delivery. In the passive condition, the odor delivery at the snout was synchronized with the first possible whisker contact with the sweeping gratings. In both stimulation conditions, the odors were delivered at a fixed time, independently of respiration.

All parts of the stimulation system were timed and synchronized with a data acquisition card (USB-6343-BNC, National Instruments) and MATLAB scripts (Mathworks). Bimodal conditions were presented 10 times (40 total), unimodal conditions 20 times (80 total), and the blanks 30 times, for a total of 150 pseudo-randomized trials with one stimulus presentation every 19 s.

**Two-photon calcium imaging in awake mice.** One week before imaging, mice were trained to stand still, head-fixed under the microscope for 5 consecutive days for 15 min to 1 h per day. Then mice were imaged for 1 h long sessions with up to four vertical depths imaged per mouse on different days. Imaging was performed using a two-photon microscope (Femtonics, Budapest, Hungary) equipped with an 8 kHz resonant scanner combined with a pulsed laser (MaiTai-DS, SpectraPhysics, Santa Clara, CA, USA) tuned at 920 nm. The objective was a ×10 Olympus (XLPLN10XSVMP), obtaining a field of view of $1000 \times 1000 \, \mu m$. Images were acquired at 31.5 Hz during trials of 11 s interleaved with an 8 s interval. For the multi-odor experiment in the passive context (Fig. 7), layer 2/3 barrel cortex two-photon imaging was performed with an acousto-optic microscope (Karthala) combined with a pulsed laser (Insight, SpectraPhysics). The objective was a 16× (N16XLWD-PF, Nikon). Images were acquired from four plans at 19.1 Hz per plan interleaved by 60 μm with fields of view of $478 \times 478 \, \mu m$.

**Calcium imaging data analysis.** Data analysis was performed using Python scripts. Motion artifacts, region of interest selection, and signal extraction were carried out using Suite2p[55]. Neuropil contamination was subtracted by applying the following equation: $F_{cor}(t) = F(t) - 0.7 \, F_n(t)$. Then the change in fluorescence $\Delta F/F_0$ was computed as $(F_{cor}(t) - F_0)/F_0$, where $F_0$ is estimated as the 8th percentile of F for each trial. To account for baseline fluctuations, the baseline activity 1–2 s before stimulus onset was subtracted for all cells in each trial. In total, 9714 neurons were recorded from 12 mice over 20 sessions in the active condition, and 14,408 neurons from 6 mice over 19 sessions in the passive condition. Among those populations, 1907 (19.6%) and 5162 (35%) neurons in the active and passive contexts, respectively, were responsive to at least one of the nine stimulation conditions (significance threshold = 5%; Kruskal–Wallis test) and were kept for analysis. Single-cell $\Delta F/F_0$ averages were computed over the 2 s of stimulus presentation and were considered odor-responsive if their mean response was significantly different in the comparison between bimodal trials with grating-only

trials or odor-only trials with blanks. For simplicity, we first evaluated the modulation by grouping together trials with odors, regardless of odor identity, resulting in four condition categories (bimodal, grating only, odor only, and blank; Figs. 2–4). We defined the olfactory modulation index as $(\Delta F/F_{odor} - \Delta F/F_{no \, odor})/\Delta F/F_{no \, odor}$ with $\Delta F/F_{odor}$ and $\Delta F/F_{no \, odor}$ the average response over stimulus presentation to bimodal and grating-only conditions or odor only and blank, respectively[56].

**Whisker tracking and breathing monitoring.** The full whisker pad was monitored with a high-speed camera (HXC20, Baumer) at 500 Hz and $608 \times 600$ pixels per frame under infrared backlight. Automated tracking of the whiskers and extraction of angle and curvature was performed with Whisk[20]. For each frame, angles and curvatures of detected whiskers (~12 per frame) were averaged to obtain a global measurement for the whisker pad. The phase of the whisking cycle was computed by applying a Hilbert transform to the band-pass filtered azimuthal angle (2nd order Butterworth filter, 4–30 Hz) from which the time point of maximum protraction and retraction were retrieved[33]. Amplitude and setpoint were then defined as the range and center of angular motion over a single cycle after quadratic interpolation. Absolute change in curvature was obtained by taking the absolute curvature and subtracting a baseline defined as the average absolute curvature one to 2 s before stimulus presentation.

Breathing was monitored with a microbridge mass air-flow sensor (Honeywell AWM3300V, Morris Plains, NJ) positioned in front of the animal's snout and perpendicular to the air stream[57]. A negative change in voltage corresponds to inhalation. Traces were sampled at 1 kHz and band-pass filtered (2nd order Butterworth filter, 4–20 Hz) and a Hilbert transform was used similarly to whisking to retrieve the breathing cycles from which breathing amplitude and frequency were computed.

**Silencing of cholinergic inputs.** For pharmacological manipulation of cholinergic signaling, 400 nl of 1 mM atropine and mecamylamine were injected with a glass micropipette and a programmable oil-based injector (Nanoliter 2020 & Micro 4; WPI) at 30 nl/min into a barrel cortex via a circular 1 mm diameter hole through the cranial window while the animal was awake. The hole was drilled with a spherical (1 mm diameter) diamond drill bit (Strauss&Co) ~1 mm away from the imaging location without breaking the glass window. The pipette was slowly inserted at 30° angle to reach the recording location at 0.5 mm depth. To allow for diffusion, we waited 30 min after injection before starting the recording. Controls were performed by injection of 400 nl of saline. Control and silencing recordings were made with a 2-day interval starting with controls first for half the experiments and silencing first for the other half. A similar plane was imaged in control and silencing sessions, but the identity of cells was not tracked across sessions.

**Stimulus decoding.** To evaluate the robustness of odor-evoked activity in the barrel cortex, we tested the accuracy at which the presented stimulus could be decoded from single-trial population responses by training and testing the nearest centroid classifier with a stratified k-fold cross-validation procedure implemented in Scikit-learn. Single-trial population vectors were constructed with neurons recorded from all sessions using $\Delta F/F$ averages in 1 s time bins. Training and testing sets were created by randomly partitioning single trials in k = 10 pairwise disjoint groups with an equal number of trials from the two tested conditions, each serving as a test set against the others. The accuracy of the classification was defined as the average performance on these 10 iterations. To evaluate its significance, the same procedure was performed for n = 1000 shuffles of the conditions and the accuracy of the non-shuffled data was located in the distribution of shuffled accuracies to obtain a P value. Note that the precision of the P value was limited to 3 decimals as 1000 shuffles were performed. For decoding between sessions (Supplementary Fig. 4), population vectors were constructed with neurons recorded in single sessions using $\Delta F/F$ averages in 1-s time bins. The same k-fold cross-validation procedure was applied. Each session was used to obtain a cross-validated accuracy with and without shuffling. Significance was assessed by comparing the mean accuracy of the real and shuffled data (Wilcoxon signed-rank test). The data were z-scored to reach significance in the results presented in Supplementary Fig. 4. The same procedure was applied for the multi-class classification presented in Fig. 7 except that linear Support Vector Machines (SVM) were used. For the categorization of amyl acetate and ethyl butyrate mixtures, we used an exponential radial basis function kernel. The results presented in Fig. 7 do not depend on the regularization parameter which was fitted for each dataset to maximize decoding at stimulus presentation while keeping performance at baseline to chance levels.

**Facial behavior analysis.** Facial movements were monitored with a CCD camera (GC651MP, Smartek Vision) equipped with a 50 mm objective (Fujinon, HF50HA-1B, Fujifilm), recording at 20 Hz. Video and two-photon acquisition were synchronized with a common trigger. Principal components were extracted using the singular value decomposition provided by the Facemap algorithm (https://github.com/MouseLand/facemap). Linear multi-class SVMs trained on the 500 first principal components of each imaging session were used to classify odor identity from behavior.

**Statistical analysis**. All quantification and statistical analysis were performed with Python scripts. Plotting relied on Matplotlib and Seaborn. Statistical assessment was performed with non-parametric tests implemented in the Statistical functions module reported in figures and legends together with mean and 95% confidence intervals, the number of samples used for the test, and the nature of the samples (number of sessions for between sessions assessment and number of trials for within sessions assessment). Hypotheses were two-sided and significance thresholds were set at 5%. Confidence intervals were computed by the bootstrap procedure implemented in Seaborn with $n = 1000$ bootstrap iterations. In all analyses, all subjects which underwent a particular protocol in the study were included.

**Reporting summary**. Further information on research design is available in the Nature Research Reporting Summary linked to this article.

## Data availability

The complete dataset supporting these findings is freely available from the Zenodo database https://doi.org/10.5281/zenodo.6397722. The data plotted in the main figures are provided in the Supplementary Information/Source Data file. Source data are provided with this paper.

## Code availability

The complete python code of the data analysis is freely available from the Zenodo database https://doi.org/10.5281/zenodo.6397722.

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

## Acknowledgements

We thank Sophie Bagur for helpful comments on the manuscript. This work was supported by the Fondation pour la Recherche Médicale (FRM grant number ECO20170637482 to A.R.), the International Human Frontier Science Program Organization (CDA-0064-2015), the European Research Council (ERC CoG 770841) and the Paris-Saclay University, the Fondation pour l'Audition, FPA IDA02 and APA 2016-03. B.B. acknowledges the support of the Fondation pour l'Audition to the Institut de l'Audition.

## Author contributions

A.R. and B.B. performed the experiments; A.R. and E.H. constructed the stimulators; A.R. analyzed the data, A.R., E.H., and B.B. designed the study and wrote the manuscript.

## Competing interests

The authors declare no competing interests.
