## [Peer Review File · Nature Communications]

REVIEWER COMMENTS

Reviewer #1 (Remarks to the Author):

Bathellier and colleagues provide convincing data that odor information reaches somatosensory (barrel) cortex in mice using 2p microscopy imaging in barrel cortex in head-fixed mice together with highly precise odor delivery and whisker stimulation (and measurement). They demonstrate that a significant fraction of neurons in barrel cortex (up to around 20%) are modulated by odor presentation. They show that part of this modulation is due to changes in whisking behavior as preventing whisking (by cutting the nerves controlling the whisker pad) reduces the modulation by odors. Intriguingly, however, even in the absence of potential odor-modulation of whisking behavior, odor information still reaches barrel cortex. Importantly, barrel cortex population activity allows decoding of odor identity (a linear classifier successfully discriminated between two different odor stimuli). Moreover, representation of odors and of somatosensory stimuli is encoded in a separable manner (as training classifiers on the condition without odors readily translated into equally high performance for test trials with odors).

The authors thus nicely show the presence of olfactory information in barrel cortex, adding to the growing body of literature demonstrating that information is represented across broad areas with traditional designations as “primary sensory specialized” etc becoming more and more outdated.

However, while the authors demonstrate this qualitative point well, it remains unclear how odor information reaches barrel cortex (seemingly not through cholinergic modulation as shown by the authors) or what this information might be good for. I would have at least expected to go substantially beyond just establishing the presence of some information and for example challenging the system with tasks of increasing complexity, assessing whether odor information is fully independent from somatosensory (as sort of suggested by Fig 6 for the simple stimuli used), whether it hinders or helps encoding of complex whisker input etc. In a similar vein (albeit maybe less pressing at this stage) the data shown for the – as highlighted by the authors - very different odor pair amyl acetate ethyl butyrate seems to indicate that olfactory information is quite corrupted in barrel cortex or at least difficult to extract with linear classifiers. In order to obtain some insight or ideas as to what this information might be good for, more challenging stimuli (e.g. the traditional binary mixtures often used including by the senior author before) or panels of odors would have to be used.

I have a few minor comments below but overall I am on the fence whether the – very convincing but somewhat basic - demonstration of the presence of some olfactory information in barrel cortex without any insight into the how or the what for (at least along the lines of more advanced stimulus sets) is enough to cross the high bar of Nature Communications.

Minor comments:

Supp Fig 1 –

“snout” not “snou”

High mag of PID trace (first 100 ms) would help to further ascertain the exquisite stimulus control

Methods: “Four injections of 200 nl of” – seems to lack a verb

Does snipping the facial nerve impact on sniff properties (inhalation length / depth) or nares movement?

The authors comment on the stationarity of the flow but I was not able to find any documentation of that stationarity

Can the authors elaborate more on how the hole in the cranial window looked like / was achieved

More details on the stimulus control would be important - I don't understand the stimulus synchronization – when exactly are odors delivered? How are they delivered relative to respiration? How is this coordinated with the time whiskers hit the grating in the active setting or how is the grating movement locked to respiration in the passive case?

I did not find the dashed line or shading mentioned in the legend to 1d?

Can the authors discuss more global attention effects – like the ones described by the Carandini / Harris lab as dominating responses across the brain. Odor presentation might result in different “attention states”, partially reflected in barrel cortex activity (I realize that the fact that odor identity can be partially decoded makes that less likely to be the only effect; on the other hand the attention effects described previously are also not necessarily just binary)

Why was breathing reduced upon odor presentation? Concentrations seem low so I would not expect a startling “holding the breath” response. I would have rather expected that – at least for initial odor presentation – sniff frequency increases?

Fig 5b – y-axis should be 10,20...50 as the axis label states“%”

Conclusion “ making it unlikely that these modulations reflect intact efference copies of whisking programs after facial nerve sectioning” – not sure I follow this logic. The odor discrimination is quite simple, it is well possible that the changes to the whisker programs would be different for the two odors. Maybe worth investigating whether advanced classifiers could detect odor identity from whisking behavior (in animals with intact whisker pad). (along the line of my initial “major” comment)

Reviewer #2 (Remarks to the Author):

Renard and colleagues address multisensory activity in barrel cortex. Specifically, they perform calcium imaging of mouse barrel cortex during olfactory and tactile stimulation. They show that ~20% layer 2/3 neurons are modulated by odors during active touch. Because odorant delivery alters whisking, they consider the possibility that this multisensory effect is due to movement. When paralyzing the whisker muscles and passively presenting tactile stimuli and/or odors, they find that ~10% of cells are odor-modulated, suggesting that odor modulation is partially due to refferent movement signals and partially due to some other mechanism. One plausible mechanism is cholinergic modulation, but pharmacological blockade of cholinergic receptors increases rather than decreases modulation. Finally, they report that they can decode odor identity from population activity in barrel cortex and that these olfactory representations are independent of the tactile representation.

The questions are interesting. The experiments appear well executed, and the datasets are impressively large.

MAJOR

1. Blockade of muscarinic and nicotinic cholinergic receptors did not produce a statistically detectable change in % of modulated cells averaged across sessions, but seemed to increase responsivity in 4(?) of the 12 sessions, potentially in a statistically significant way (Fig.5). It would sense to test the individual sessions as well. The cholinergic blocker results appear mixed. If there are positive effects there, the

authors' conclusion that acetylcholine is not responsible for the olfactory effect still stands. Nevertheless, there is no reason to ignore other effects in the data.

2. A key analysis for the paper's conclusion appears in Fig.6d which shows that odor identity can sometimes be decoded from barrel cortex activity. The authors interpret this to mean that barrel cortex contains odorant-specific information. A trivial alternative explanation for this result would be if one odorant is more strongly driving barrel cortex activity than the other odorant is. Analyses to examine whether or not this is the case are needed (e.g., mean population responses for each odorant, a scatterplot of response magnitude for one odor on the X-axis and that of the other on the Y-axis, etc).

3. Fig.6f (grating decoding accuracy with and without odors) is consistent with the authors' idea that olfactory and tactile representations are orthogonal in barrel cortex. A limitation of this analysis is that the tactile representation is very strong in barrel cortex (Fig.6c), and the analysis is only binary. A stronger piece of evidence would be if the accuracy of odor identity decoding were also unchanged by tactile stimulation. The authors could simply perform a similar analysis to Fig.6f for this.

4. Fig.6h (the dot product of the coding vectors) is supportive evidence of the orthogonality conclusion, but could also be analyzed further. How is it that the dot product of the decoding vectors comes to be 0? One simple explanation could be that while some neurons are multimodal there are many neurons that are unimodal (the representations could be much more separate than the cartoons would suggest). An analysis that could address would be to recompute the dot product for various subsets of the data (the multimodal cells only, touch-only cells + odor-only cells). On a related note, the earlier sections of the Results should break down the proportion of cells respond to touch-only, odor-only, both, and neither for the passive and active designs. These basic numbers seem to be missing.

5. More details about the sensory stimulation should be provided. For instance, the Methods should report the distance to the mouse's face where the gratings are presented for both active and passive stimulation. The speed and duration of the grating sweeps during passive presentation and the number of direction changes should be clearly stated. The speed at which the linear actuator advances the grating should also be given. Does the top panel in Fig.1d illustrate the grating as it is moving or after it comes to a rest in its most extended position?

MINOR

1. Scale in Fig.1d video?

2. The end of Results re Fig.2 (active) explicitly says that increases in activity during blank trials are due to whisking, but that can't be true as the experiments go on to show. In Fig.1 (passive), Cell 1 responds to a double blank. What is driving that?

3. Fig.3b, bottom. Are "Grating only" or "Blank" trials plotted as suggested by the inset? I don't see these colors in the scatterplots.

4. The Supp.Fig.3 legend needs a few words to explain the difference between the global and local analyses, which the reader has not yet seen in the Results the first time this figure is called out.

5. The Results refer to Fig.6e as if it is real data, but I think it's just a cartoon. I suspect the references to Fig.6e and 6f were accidentally switched.

6. "Alternatively, barrel cortex receives inputs from several thalamic nuclei, including from secondary thalamic regions which themselves are known to receive multimodal inputs⁵²." This seems the wrong reference (primate) for a statement about barrel cortex. Is this meant?

Reviewer #3 (Remarks to the Author):

This study investigates whether olfaction and whisker sensation interact in the brain, and presents some interesting, novel findings. The principal claims are (1) that odors alter barrel cortex activity and (2) that they do so via at least two mechanisms - first by enhancing whisking, second by "central cross-talk". Further claims are that the odor modulation can both enhance and suppress activity, and that odor identity and tactile stimuli can be decoded independently from the barrel cortex activity.

The claims are interesting and of potential wide interest in systems neuroscience.

The data in the study support claim 1 (but see below). Concerning claim 2, it is convincing that odor-modulation is at least partly due to odor-enhancement of whisking. The idea is that odors elicit an exploratory response where mouse whisking is increased, thus causing a refferent increase in ascending sensory drive to the barrel cortex. This is interesting since it shows that, in behaving animals, there is cross-talk between modalities that the brain must somehow deal with in order to achieve

reliable behavior (the authors' decoding analysis speaks to this). The authors also claim evidence for "central cross-talk". This is potentially even more interesting – since it suggests novel mechanisms – but further work is needed to strengthen this conclusion.

Major

- The authors report that the fraction of cells that are odor-modulated is 21.7% in the active context. However, the data in Fig 2d raise some questions – there are non-modulated cells that deviate as much from the main diagonal as modulated cells. These results are based on statistical comparisons using a 5% significance level, which implies that the 21.7% figure is inflated upwards by false positives. The authors' shuffling test (Fig 2e) makes it convincing that there is a genuine effect of odor-modulation, but the fraction of odor-modulated cells should be more conservatively/accurately determined.
- The authors report that 8.99%/10.81% of cells are odor-modulated in the passive context. Since the passive context is designed to eliminate whisker movement, this is the critical evidence for a mechanism beyond odor-enhancement of whisking being involved, and is therefore a key point of the paper. My concern is that these 9-11% figures are likely to be inflated by false positives and are not far off the 5% level expected by chance. First, accurate (unbiased) figures for the fraction of odor-modulated cells should be determined. Second, since the real fraction of odor-modulated cells in the passive context is low (taking into account false positives, it is likely ~5%), can the authors add any further evidence to exclude that the result might be due to an artefact (eg incomplete nerve sectioning or residual movement in non-tracked whiskers)?
- In Figure 5b, there are signs of the test between the green bars being underpowered. What is the sample size for the test? It is not clear that it is safe to infer from the negative test outcome that "cholinergic signaling is not one of the mechanisms". A safer inference is that no evidence for a cholinergic mechanism was found.

Minor

- The authors show interesting evidence for the representations of tactile grating and odor being independent. However, it is not clear whether these data are for the active or passive contexts. Given that the passive removes the whisking component it would be interesting to see results for both contexts.

Reviewer #1 (Remarks to the Author):

Bathellier and colleagues provide convincing data that odor information reaches somatosensory (barrel) cortex in mice using 2p microscopy imaging in barrel cortex in head-fixed mice together with highly precise odor delivery and whisker stimulation (and measurement). They demonstrate that a significant fraction of neurons in barrel cortex (up to around 20%) are modulated by odor presentation. They show that part of this modulation is due to changes in whisking behavior as preventing whisking (by cutting the nerves controlling the whisker pad) reduces the modulation by odors. Intriguingly, however, even in the absence of potential odor-modulation of whisking behavior, odor information still reaches barrel cortex. Importantly, barrel cortex population activity allows decoding of odor identity (a linear classifier successfully discriminated between two different odor stimuli). Moreover, representation of odors and of somatosensory stimuli is encoded in a separable manner (as training classifiers on the condition without odors readily translated into equally high performance for test trials with odors).

The authors thus nicely show the presence of olfactory information in barrel cortex, adding to the growing body of literature demonstrating that information is represented across broad areas with traditional designations as “primary sensory specialized” etc becoming more and more outdated.

However, while the authors demonstrate this qualitative point well, it remains unclear how odor information reaches barrel cortex (seemingly not through cholinergic modulation as shown by the authors) or what this information might be good for. I would have at least expected to go substantially beyond just establishing the presence of some information and for example challenging the system with tasks of increasing complexity, assessing whether odor information is fully independent from somatosensory (as sort of suggested by Fig 6 for the simple stimuli used), whether it hinders or helps encoding of complex whisker input etc. In a similar vein (albeit maybe less pressing at this stage) the data shown for the – as highlighted by the authors - very different odor pair amyl acetate ethyl butyrate seems to indicate that olfactory information is quite corrupted in barrel cortex or at least difficult to extract with linear classifiers. In order to obtain some insight or ideas as to what this information might be good for, more challenging stimuli (e.g. the traditional binary mixtures often used including by the senior author before) or panels of odors would have to be used.

I have a few minor comments below but overall I am on the fence whether the – very convincing but somewhat basic - demonstration of the presence of some olfactory information in barrel cortex without any insight into the how or the what for (at least along the lines of more advanced stimulus sets) is enough to cross the high bar of Nature Communications.

We thank the referee for his/her thorough examination of the manuscript and the frank evaluation of the impact. In order to take this into account, we have performed two types of experiments:

First, we searched for the potential source of olfactory information that is present into the barrel cortex. For this we have performed chemogenetic inactivation of the perirhinal cortex during olfactory-tactile stimulation in the awake mouse. Perirhinal cortex is one of the

few cortical areas that receives strong inputs from the piriform cortex and strongly projects to barrel cortex. We used chemogenetic manipulations due to the large antero-posterior extent of perirhinal cortex which makes it difficult to ensure extensive light delivery for optogenetics, and due to its narrow dorso-lateral extent which makes muscimol silencing difficult to control. Multiple targeted AAV injections for Hm4di expression provided sufficient precision. Systemic CNO injections failed to reduce olfactory responses (see figures below this comment). Given the difficulty of interpreting this negative result (e.g. Was HM4DI expression broad enough? Was CNO diffusion to the target efficient ?), we decided not to include it in the manuscript.

In addition, we strongly reinforced the demonstration that genuine and precise olfactory information reaches the barrel cortex. We did that by imaging S1 during the presentation of 5 different odors, as well as odor mixtures. The results, presented in a new figure (Fig 7), show that odor identity for these diverse stimuli can be decoded above chance level and that S1 representations respect the continuity of mixtures. In addition, we have shown using classifiers that neither breathing nor facial activity modulations can provide comparable levels of information about odor identity. Together, this strongly suggests that olfactory information in the barrel cortex represents usable sensory information instead of a correlate of behavioral modulations. We believe that, with these demonstrations, the message of our manuscript is substantially strengthened and hope that it now reaches the bar for Nature Communications.

Figure: Odor-induced activity in barrel cortex during perirhinal chemogenetic silencing. **a** Averaged population response to bimodal trials for the saline (blue) and CNO (orange) conditions. Shadings indicate 95% c.i. within sessions. **b.** Proportion of modulated cells for the saline (blue) and CNO (orange) conditions. Shadings indicate 95% c.i. across sessions. **c-d** Same as a-b for olfactory stimulations alone.

Figure: Decoding of olfactory information in the barrel cortex during perirhinal silencing. **a** Decoding accuracy of centroid classifiers of bimodal and grating alone trials from wS1 activity averaged over 1 sec time bins with the global population of cells recorded over all sessions for the saline (blue) and CNO (orange) conditions. Performance for shuffled labels in saline condition is shown in gray. Error bars indicate 95% c.i. with $n = 10$ for real data and $n = 100$ for shuffled data; each data point is the average of a 20-fold stratified cross-validation; shading indicates stimulus presentation. **b-d** As in a for decoding of odor alone and blank trials, grating

orientation and odor identity, respectively. **e-f** As in a-d with local populations, with activity averaged between 1 and 2 sec after stimulus onset.

Minor comments:

R1.1 Supp Fig 1 –
“snout” not “snou”

Corrected.

R1.2 High mag of PID trace (first 100 ms) would help to further ascertain the exquisite stimulus control

Corrected. see Suppl. Fig. 1 b (copied below).

R1.3 Methods: “Four injections of 200 nl of” – seems to lack a verb

Corrected.

R1.4 Does snipping the facial nerve impact on sniff properties (inhalation length / depth) or nares movement?

We quantified sniffing in passive animals; see Suppl. Fig. 2 b-e (copied below). The comparison with nerve intact animals shows that breathing is not affected by facial nerve sectioning, neither at baseline nor during stimulus presentation. The non-significant increase in breathing frequency during stimulus presentation with touch inputs (panel e, right, orange and blue) indicates that the passive touch stimulation, where oriented gratings are brushed against the whiskers, slightly increases breathing frequency compared to the active touch condition.

In our initial dataset, we cannot access nostril movements because the snout is confined in the odor delivery tube. However, we recorded videos of the facial movement (see Methods) in two mice where the snout was not confined. We measured the motion energy of the snout, defined as the square of the pixel value difference between consecutive frames, before and after sectioning of the facial nerve and found that snout and nostril movements were abolished; see Suppl. Fig. 2 h (copied below).

R1.5 The authors comment on the stationarity of the flow but I was not able to find any documentation of that stationarity

We added a plot of air flow measurement during the stimulus presentation to Suppl. Fig. 1 (copied below). Blue trace represents mean over odor presentations for all pure odors and mixtures used, including the control. Gray traces correspond to single traces with $n=10$ for each stimulus. Shading indicates odor presentation. Flow change at 0.5 and 7 sec correspond to flow adjustment at opening and closing of the two odor delivery MFC's, unspecific of odor presence and identity. Note that this change in air flow is three orders of magnitude smaller than the change elicited by a sniff which is about 1 V (e.g. see breathing trace in Fig. 7d for comparison). Note also that the solenoid valve switch at stimulus onset does not disrupt the air flow. These measurements were performed with the microbridge mass air flow sensor (Honeywell AWM3300V, Morris Plains, NJ) used for respiration monitoring.

R1.6 Can the authors elaborate more on how the hole in the cranial window looked like / was achieved

We updated the methods with: “For pharmacological manipulation of cholinergic signaling, 400 nl of 1 mM atropine and mecamylamine was injected with a glass micropipette and a programmable oil-based injector (Nanoliter 2020 & Micro 4; WPI) at 30nL/min into barrel cortex via a circular 0.5 mm diameter hole through the cranial window while the animal was awake. The hole was drilled with a spherical (1 mm diameter) diamond drill bit (Strauss & Co) ~ 1 mm away from the imaging location without breaking the glass window. The pipette was slowly inserted at a 30° angle to reach the recording location at 0.5mm depth.”

R1.7 More details on the stimulus control would be important - I don't understand the stimulus synchronization – when exactly are odors delivered? How are they delivered relative to respiration? How is this coordinated with the time whiskers hit the grating in the active setting or how is the grating movement locked to respiration in the passive case?

More details about the sensory stimulation should be provided. For instance, the Methods should report the distance to the mouse's face where the gratings are presented for both active and passive stimulation. The speed and duration of the grating sweeps during passive presentation and the number of direction changes should be clearly stated. The speed at which the linear actuator advances the grating should also be given. Does the top panel in Fig.1d illustrate the grating as it is moving or after it comes to a rest in its most extended position?

We updated the method with the requested details:

“Orientated gratings were made of five ridges of 3.5 mm thickness and spacing on a 3d printed PLA disk of 35 mm diameter. Gratings were actuated with a custom-made presentation wheel consisting of two stepper motors (42BYG, Makeblock) mounted on a linear stage (eTrack, Newmark) and controlled with an Arduino. In the active condition, the gratings were brought from their starting location to the tips of the whiskers 4 cm away, reaching their most extended position after 1 sec. In the passive condition, the gratings were brought by the linear stage at the same speed, rotated 10° anterior to the mouse. Once the linear stage reached its final position, the gratings were rotated 30° forward at a speed of 35 cm/sec (thus sweeping over the whiskers in 100 ms), then rotated 30° backward 500 ms after initiation of the first sweep.”

In the active condition, the odor delivery onset at the snout location (measured with a PID; 44 ms latency after shuffle valve switch) was synchronized with the moment where the gratings reached their most extended position. As mice in the active condition tend to whisk to fetch the approaching grating cued by the sound of the linear stage, whiskers could make a first contact with the gratings up to 400 ms before odor delivery. In the passive condition, the odor delivery at the snout was synchronized with the first possible whisker contact with the sweeping gratings. In both stimulation conditions, the odors were delivered at a fixed time, independently of respiration.”

R1.8 I did not find the dashed line or shading mentioned in the legend to 1d?

The dashed lines and shading were added to Fig. 1d.

R1.9 Can the authors discuss more global attention effects – like the ones described by the Carandini / Harris lab as dominating responses across the brain. Odor presentation might result in different “attention states”, partially reflected in barrel cortex activity (I realize that the fact that odor identity can be partially decoded makes that less likely to be the only effect; on the other hand the attention effects described previously are also not necessarily just binary)

To address this important point, we performed new S1 imaging sessions with a larger set of odors (5 different molecules and one odor mixture series of amyl acetate (AA) and ethyl butyrate (EB)). Classifiers based on S1 activity reached 36% accuracy on the identification of odor identity (chance level = 20%) whereas classifiers based on extensive facial movement analysis (singular value decomposition of facial movies as in Stringer et al. 2019 from the Carandini and Harris lab) did not perform above chance level. Breathing amplitude and frequency were not modulated by odor identity (Supplementary Fig. 5). Moreover, classifiers were able to assign responses to intermediate mixtures of AA and EB to the dominant component. These results are now displayed in Fig. 7. Based on this series of results, we establish that odor-related activity in S1 represents rich olfactory information and is unlikely to reflect mere attentional modulation, at least as far as they are correlated to the animal’s facial behavior.

R1.10 Why was breathing reduced upon odor presentation? Concentrations seem low so I would not expect a startling “holding the breath” response. I would have rather expected that – at least for initial odor presentation – sniff frequency increases?

We provided a plot of breathing amplitude and frequency as a function of trial number (see Suppl. Fig. 2f, g ; copied below). Sniffing frequency is ~0.5-1 Hz higher at the beginning of the session. In breathing frequency that drops after the initial odor presentations. This effect is likely reduced by the fact that mice were exposed to 10 odor stimulations before starting each recording session.

We now compute breathing frequency more directly by counting the breathing cycles obtained by applying a Hilbert transform to the breathing signal (cf. Methods), rather than using the peak in the power spectrum computed with Welch’s method, which resulted in a discrete measure due to the short 2 sec signal on which it was applied. As a result, the impact of odors on breathing frequency (Fig. 3f) is no longer significant. However, a tendency towards a reduction in breathing frequency was observed (Fig. 3f, bottom). Even though odor concentrations are low, trials where this is observed, such as the example presented below, do indeed suggest a “holding of breath”, likely due to the absence of a behavioral task in our protocol.

We updated the text (page 10) with: “As there is an intricate link between whisking behavior and sniffing, we also checked if breathing was impacted by odors. Breathing amplitude (Fig. 3e), measured in a 2 sec time window starting at odor onset, was significantly reduced in the presence of odors compared to the conditions without odors. A similar tendency could be found in breathing frequency (Fig. 3f), but the reduction was not robust enough to be significant.”

R1.12 Fig 5b – y-axis should be 10,20...50 as the axis label states“%”

Corrected.

R1.13 Conclusion “ making it unlikely that these modulations reflect intact efference copies of whisking programs after facial nerve sectioning” – not sure I follow this logic. The odor discrimination is quite simple, it is well possible that the changes to the whisker programs would be different for the two odors. Maybe worth investigating whether advanced classifiers could detect odor identity from whisking behavior (in animals with intact whisker pad). (along the line of my initial “major” comment)

To support this claim we have now shown:

- that odor discrimination can be performed above chance for 5 different odors (Fig. 7c-d)
- that residual whisker movements and facial movement are not predictive of odor identity for the 5 odors (Fig. 7h)

- that breathing signal in the nerve-sectioned condition does not predict odor identity (Supplementary Fig. 5).
- We used SVMs on whisking data in intact whisker pad conditions and found that whisker movements do not predict odor identity (Supplementary Fig. 5).

Reviewer #2 (Remarks to the Author):

Renard and colleagues address multisensory activity in barrel cortex. Specifically, they perform calcium imaging of mouse barrel cortex during olfactory and tactile stimulation. They show that ~20% layer 2/3 neurons are modulated by odors during active touch. Because odorant delivery alters whisking, they consider the possibility that this multisensory effect is due to movement. When paralyzing the whisker muscles and passively presenting tactile stimuli and/or odors, they find that ~10% of cells are odor-modulated, suggesting that odor modulation is partially due to refferent movement signals and partially due to some other mechanism. One plausible mechanism is cholinergic modulation, but pharmacological blockade of cholinergic receptors increases rather than decreases modulation. Finally, they report that they can decode odor identity from population activity in barrel cortex and that these olfactory representations are independent of the tactile representation. The questions are interesting. The experiments appear well executed, and the datasets are impressively large.

MAJOR

R2.1 Blockade of muscarinic and nicotinic cholinergic receptors did not produce a statistically detectable change in % of modulated cells averaged across sessions, but seemed to increase responsivity in 4(?) of the 12 sessions, potentially in a statistically significant way (Fig.5). It would sense to test the individual sessions as well. The cholinergic blocker results appear mixed. If there are positive effects there, the authors' conclusion that acetylcholine is not responsible for the olfactory effect still stands. Nevertheless, there is no reason to ignore other effects in the data.

We tested the significance of the change in the fraction of odor-responsive cells in each session individually using a Chi-square test. (Note that due to noise correlations, statistical independence of recorded neurons in a session is not guaranteed, while it is an assumption of the test). Significant sessions are now represented by a thicker black line. Assessed this way, the proportion of modulated cells was found to decrease significantly in a single session both when comparing the effect of odors with and without touch, while one session showed a significant increase in the absence of touch. Therefore, we maintain the conclusion that “we found no evidence that cholinergic signaling underlies odor-driven activity.”

R2.2 A key analysis for the paper's conclusion appears in Fig.6d which shows that odor identity can sometimes be decoded from barrel cortex activity. The authors interpret this to mean that barrel cortex contains odorant-specific information. A trivial alternative explanation for this result would be if one odorant is more strongly driving barrel cortex activity than the

other odorant is. Analyses to examine whether or not this is the case are needed (e.g., mean population responses for each odorant, a scatterplot of response magnitude for one odor on the X-axis and that of the other on the Y-axis, etc).

We plotted the mean population response for each odorant as requested (Supplementary Fig. 5; copied below). The odorants elicited the same level of population activity in all three datasets.

Moreover, we now also show in Fig. 7 that odor identity of 5 different odors can be decoded from barrel cortex activity, in line with the presence of odor identity encoding cells.

R2.3 Fig.6f (grating decoding accuracy with and without odors) is consistent with the authors' idea that olfactory and tactile representations are orthogonal in barrel cortex. A limitation of this analysis is that the tactile representation is very strong in barrel cortex (Fig.6c), and the analysis is only binary. A stronger piece of evidence would be if the accuracy of odor identity decoding were also unchanged by tactile stimulation. The authors could simply perform a similar analysis to Fig.6f for this.

We performed the same analysis as Fig. 6f (added to Fig. 6; copied below) to test if the accuracy of odor identity decoding was unchanged by tactile input. While tactile information is not affected by the presence of odors, the decoding of odor identity is impacted by tactile inputs such that discrimination goes back to chance level. We thank the referee for pointing out this analysis which shows that tactile and odor representations are not fully orthogonal. It seems that because tactile representations are based on stronger activity than odor representations, they are little affected by them. Odor representations, however, depend on the tactile input. We corrected this in the manuscript and removed the orthogonality analysis which was flawed, as we now realized, by the fact that the norm of odor representations is very small (see next comment).

R2.4 Fig.6h (the dot product of the coding vectors) is supportive evidence of the orthogonality conclusion, but could also be analyzed further. How is it that the dot product of

the decoding vectors comes to be 0? One simple explanation could be that while some neurons are multimodal there are many neurons that are unimodal (the representations could be much more separate than the cartoons would suggest). An analysis that could address would be to recompute the dot product for various subsets of the data (the multimodal cells only, touch-only cells + odor-only cells). On a related note, the earlier sections of the Results should break down the proportion of cells that respond to touch-only, odor-only, both, and neither for the passive and active designs. These basic numbers seem to be missing.

Based on the comment above we removed the orthogonality analysis. Nevertheless, we have performed it on different subsets of cells so that the referee can appreciate that it was biased by the large sample size and the fact that odor representations are very diluted. When we focus on cells that respond to odor or touch (Figure below; right) there is no orthogonality.

Dot product of grating orientation and odor identity decoding vectors (black). The scalar product was computed between the odor identity decoding vector across time and the grating orientation decoding vector fixed over the stimulus presentation. The same operation between the grating orientation decoding vector and itself is shown for comparison (blue). Shading indicates 95% CI over 10 random 2-fold train/test splits. **a** All active cells. **b** cells that respond to tactile stimuli and whose response is modulated by odors. **c** Cells that respond to odor or touch.

We have also now reported the fraction of touch responsive cells in the first results paragraph. The fraction of odor responsive cells were given in the second paragraph and the fraction of unresponsive cells were given in the first paragraph as one minus the fraction of cells responding to any stimulus.

R2.5 More details about the sensory stimulation should be provided. For instance, the Methods should report the distance to the mouse's face where the gratings are presented for both active and passive stimulation. The speed and duration of the grating sweeps during passive presentation and the number of direction changes should be clearly stated. The speed at which the linear actuator advances the grating should also be given. Does the top panel in Fig.1d illustrate the grating as it is moving or after it comes to a rest in its most extended position?

We specified in the legend of Fig. 1d that the image corresponds to the grating at its most extended position. We updated the methods with the requested details:

"Orientated gratings were made of five ridges of 3.5 mm thickness and spacing on a 3d printed PLA disk of 35 mm diameter. Gratings were actuated with a custom-made

presentation wheel consisting of two stepper motors (42BYG, Makeblock) mounted on a linear stage (eTrack, Newmark) and controlled with an Arduino. In the active condition, the gratings were brought from their starting location to the tips of the whiskers 4 cm away, reaching their most extended position after 1 sec. In the passive condition, the gratings were brought by the linear stage at the same speed, rotated 10° anterior to the mouse. Once the linear stage reached its final position, the gratings were rotated 30° forward at a speed of 35 cm/sec (thus sweeping over the whiskers in 100 ms), then rotated 30° backward 500 ms after initiation of the first sweep.

In the active condition, the odor delivery onset at the snout location (measured with a PID; 44 ms latency after shuffle valve switch) was synchronized with the moment where the gratings reached their most extended position. As mice in the active condition tend to whisk to fetch the approaching grating cued by the sound of the linear stage, whiskers could make a first contact with the gratings up to 400 ms before odor delivery. In the passive condition, the odor delivery at the snout was synchronized with the first possible whisker contact with the sweeping gratings. In both stimulation conditions, the odors were delivered at a fixed time, independently of respiration.”

MINOR

R2.6 Scale in Fig.1d video?

Corrected..

R2.7 The end of Results re Fig.2 (active) explicitly says that increases in activity during blank trials are due to whisking, but that can't be true as the experiments go on to show. In Fig.1 (passive), Cell 1 responds to a double blank. What is driving that?

Actually in Fig. 1 there are no examples from the passive context. We clarified this in the caption. So the response seen in Fig. 1 Cell 1 is likely due to whisking.

R2.8 Fig.3b, bottom. Are “Grating only” or “Blank” trials plotted as suggested by the inset? I don't see these colors in the scatterplots.

We clarified the legend.

R2.9 The Supp.Fig.3 legend needs a few words to explain the difference between the global and local analyses, which the reader has not yet seen in the Results the first time this figure is called out.

We clarified this by changing the terminology in the figure to match that used in the text.

R2.10 The Results refer to Fig.6e as if it is real data, but I think it's just a cartoon. I suspect the references to Fig.6e and 6f were accidentally switched.

We corrected the reference.

R2.11 “Alternatively, barrel cortex receives inputs from several thalamic nuclei, including from secondary thalamic regions which themselves are known to receive multimodal inputs⁵².” This seems the wrong reference (primate) for a statement about barrel cortex. Is this meant?

We changed this reference to a broader review of the thalamic influences on multisensory perception which includes evidence gathered on rodents.

Tyll, S., Budinger, E. & Noesselt, T. Thalamic influences on multisensory integration. *Commun. Integr. Biol.* **4**, 378–381 (2011).

Reviewer #3 (Remarks to the Author):

This study investigates whether olfaction and whisker sensation interact in the brain, and presents some interesting, novel findings. The principal claims are (1) that odors alter barrel cortex activity and (2) that they do so via at least two mechanisms - first by enhancing whisking, second by “central cross-talk”. Further claims are that the odor modulation can both enhance and suppress activity, and that odor identity and tactile stimuli can be decoded independently from the barrel cortex activity.

The claims are interesting and of potential wide interest in systems neuroscience.

The data in the study support claim 1 (but see below). Concerning claim 2, it is convincing that odor-modulation is at least partly due to odor-enhancement of whisking. The idea is that odors elicit an exploratory response where mouse whisking is increased, thus causing a reafferent increase in ascending sensory drive to the barrel cortex. This is interesting since it shows that, in behaving animals, there is cross-talk between modalities that the brain must somehow deal with in order to achieve reliable behavior (the authors’ decoding analysis speaks to this). The authors also claim evidence for “central cross-talk”. This is potentially even more interesting – since it suggests novel mechanisms – but further work is needed to strengthen this conclusion.

Major

R3.1 The authors report that the fraction of cells that are odor-modulated is 21.7% in the active context. However, the data in Fig 2d raise some questions – there are non-modulated cells that deviate as much from the main diagonal as modulated cells. These results are based on statistical comparisons using a 5% significance level, which implies that the 21.7% figure is inflated upwards by false positives. The authors’ shuffling test (Fig 2e) makes it convincing that there is a genuine effect of odor-modulation, but the fraction of odor-modulated cells should be more conservatively/accurately determined.

The authors report that 8.99%/10.81% of cells are odor-modulated in the passive context. Since the passive context is designed to eliminate whisker movement, this is the critical evidence for a mechanism beyond odor-enhancement of whisking being involved, and is therefore a key point of the paper. My concern is that these 9-11% figures are likely to be inflated by false positives and are not far off the 5% level expected by chance. First, accurate (unbiased) figures for the fraction of odor-modulated cells should be determined. Second, since the real fraction of odor-modulated cells in the passive context is low (taking into account false positives, it is likely ~5%), can the authors add any further evidence to exclude that the result might be due to an artefact (eg incomplete nerve sectioning or residual movement in non-tracked whiskers)?

The reviewer is right that the reported fraction is inflated by false positive rates, as is usually the case with data that has a similar level of signal and noise. If we take a more conservative threshold at $\alpha=0.01$, the fractions go down to 2,2 and 2,6%, and for $\alpha = 0,005$ they shift to 1,3 et 1.4%. The test detects consistently about twice the number of cells one would expect by chance, indicating that odor-induced changes in neuronal activity are real but that, because they are close to noise levels, it is not possible to unambiguously discriminate on a statistical basis the true and false positives, whatever the alpha value or

the scope of the test (e.g. multiple comparison corrections) may be. Hence, we are not able to provide an unbiased value for the fraction of modulated cells. Note also that probably, there is a continuum of odor modulation amplitudes, rather than two distinct populations (odor-modulated or not).

Yet we have always been concerned by this issue and this is the reason why we used cross-validated decoders of population activity to determine in an unbiased manner if there is sufficient information to decode odor presence. Results shown in Fig. 6 clearly establish this point. We also reinforced this result with new experiments with 5 odors, which provided clear examples of odor-specific cells (Fig. 7b) and further demonstrated the presence of odor information in the neuronal population. This is now clarified in the results:

“While the significant proportion of neurons that can be detected as odor-modulated suggests the presence of odor information in the barrel cortex, the intrinsic uncertainty about whether or not a given odor-modulation is a true or false detection makes the quantification of the actual level of information difficult. We therefore evaluated the information carried by odor-related activity in barrel cortex, in the passive context when whisking is abolished, using centroid classifiers with a stratified 20-fold cross-validation, trained on population activity vectors to discriminate various stimulation conditions. This approach is independent of statistical thresholds as it considers all neurons whether or not their modulation by the odors is statistically detectable.”

Moreover, in order to address the referee’s issue about incomplete nerve section or whisker tracking, we provide a new analysis in which we assess the amount of odor information in facial movements (whiskers included) after nerve sectioning (Fig. 7a). The results are provided in Fig. 7h and clearly show that no odor information is present in facial movements. This together strongly reinforces the idea that odor responses in S1 are not statistical or experimental artifacts.

R3.2 In Figure 5b, there are signs of the test between the green bars being underpowered. What is the sample size for the test? It is not clear that it is safe to infer from the negative test outcome that “cholinergic signaling is not one of the mechanisms”. A safer inference is that no evidence for a cholinergic mechanism was found.

The number of independent samples (note that cells recorded in the same session are not strictly independent because of noise correlations) is 6 (now visible in the figure), corresponding to the number of sessions. We used a chi-square test to assess whether the effect measured in single sessions is significant (see response to Referee 2) and found that the few significant single session effects could be either positive or negative. (Note however that the sample independence hypothesis is not 100% fulfilled in this test). We agree with the reviewer that the lack of significance in a test is no evidence of a lack of effect and therefore we changed the conclusion of this paragraph to : “we found no evidence that cholinergic signaling underlies odor-driven activity. ”

Minor

R3.4 The authors show interesting evidence for the representations of tactile grating and odor being independent. However, it is not clear whether these data are for the active or passive contexts. Given that the passive removes the whisking component, it would be interesting to see results for both contexts.

We clarified that the analyses done in Fig. 6 were performed for data in the passive context. See page 16: “We then evaluated the type of information carried by odor-related activity in barrel cortex and to what extent this information interacts with tactile representations in the passive context where whisking is abolished.”

We also added as a supplementary figure (Supplementary fig. 6; copied below) the same analysis for the active context. In this context, the tactile information is moderately affected by odors (the decoding accuracy drops from 68.7% without odors to 59% with odors), which is likely due to the impact of odors on whisking behavior documented in Fig. 3. Similarly to the analysis done in the passive context, odor representations are modified by tactile inputs (cf. response to R2 comment 3). Note that odor decoding is not better in the active than in the passive context.

REVIEWERS' COMMENTS

Reviewer #1 (Remarks to the Author):

The authors have performed a very nice set of additional experiments. I appreciate that the (clever, complex, difficult) chemogenetic silencing experiments of perirhinal are not necessarily useful to put in the manuscript. Nevertheless I do appreciate the work that went into those.

The 5 odor experiments and the FaceMap experiments / analyses are particularly compelling, demonstrating that the information represented in barrel cortex is indeed not "only" reflecting altered behavior and does represent odors in a non-binary way.

I have no further major comments to the scientific content and find the manuscript a very valuable addition, in principle suitable for Nature Communications.

My only remaining major comment relates to data availability: The datasets are of high quality and value to the community. Before publication of the manuscript I would expect to see them somewhat curated and deposited to a repository that allows others to make use of them as well (e.g. figshare). Ideally this would be combined with e.g. a Python notebook to allow reproducing the figures but that might be a hard ask.

Minor comments:

- * There are a few small typos, commas missing, whitespaces before commas etc that deserve attention
- * on P16 "during silencing of cholinergic "  word missing

Reviewer #2 (Remarks to the Author):

The authors have adequately addressed almost all of my previous comments. They have removed their claim of orthogonal tactile and odor representations and substantiated their other claims.

My previous major comment 4 asked about the breakdown of cell response types. The authors added some percentages to the beginning of the Results, which helps. What is missing though is the % that responded to both odor and touch. Is it not possible to classify each cell as having an odor response or

touch response? It looks like the authors are only stating touch responses and total responses and asking the reader to infer the % of those that respond only to odor. Moreover, there is no indication which percentage are bimodal. My apologies if I am simply missing this, but it seems an important issue that should be explicit.

My remaining comments are all relatively minor:

Fig.6 legend needs to be updated to explain what panel c represents

Fig.6f panel title and the X-axis labels appear to be mislabeled. According to the legend, the panel plots with/without touch.

Are the parentheses in the Fig.7f inset meant to indicate that the authors have quantitatively compared these PCs with the stated variables and there is a strong correlation? If so, I think the authors should plot this somewhere in the main or supplementary figures. If not, maybe the authors should remove the parentheses and just explicitly write in the Results that these PCs appeared to the experimenters to roughly capture X, Y, and Z. I think all we can say about Fig.7f, as is, is that the PCs do not resemble neural activity.

Fig.7h. Is there enough data to perform decoder analysis with 500 PCs? If not, it might make sense to double check with less PCs (e.g., something like the first 10-100).

Reviewer #1 (Remarks to the Author):

The authors have performed a very nice set of additional experiments. I appreciate that the (clever, complex, difficult) chemogenetic silencing experiments of perirhinal are not necessarily useful to put in the manuscript. Nevertheless I do appreciate the work that went into those.

The 5 odor experiments and the FaceMap experiments / analyses are particularly compelling, demonstrating that the information represented in barrel cortex is indeed not "only" reflecting altered behavior and does represent odors in a non-binary way.

I have no further major comments to the scientific content and find the manuscript a very valuable addition, in principle suitable for Nature Communications.

R1.1 My only remaining major comment relates to data availability: The datasets are of high quality and value to the community. Before publication of the manuscript I would expect to see them somewhat curated and deposited to a repository that allows others to make use of them as well (e.g. figshare). Ideally this would be combined with e.g. a Python notebook to allow reproducing the figures but that might be a hard ask.

We have created a public repository on Zenodo with a curated dataset and Python code sufficient to reproduce the figures (<https://doi.org/10.5281/zenodo.6397722>), which is referred to in the Data and code availability section:

“Data and code availability

The complete dataset and Python analysis code supporting these findings are freely available from the open access Zenodo database <https://doi.org/10.5281/zenodo.6397722>.”

Minor comments:

R1.2 There are a few small typos, commas missing, whitespaces before commas etc that deserve attention.

We have proofread the manuscript.

R1.3 On P16 "during silencing of cholinergic "  word missing.

We inserted the missing word "inputs".

Reviewer #2 (Remarks to the Author):

The authors have adequately addressed almost all of my previous comments. They have removed their claim of orthogonal tactile and odor representations and substantiated their other claims.

R2.1 My previous major comment 4 asked about the breakdown of cell response types. The authors added some percentages to the beginning of the Results, which helps. What is missing though is the % that responded to both odor and touch. Is it not possible to classify

each cell as having an odor response or touch response? It looks like the authors are only stating touch responses and total responses and asking the reader to infer the % of those that respond only to odor. Moreover, there is no indication which percentage are bimodal. My apologies if I am simply missing this, but it seems an important issue that should be explicit.

We classified cells in “touch only”, “odor only” and “touch and odor” cells as requested; see **Supplementary Fig. 2** (copied below) and in the first subsection of the results section. These categories were defined by testing responses to tactile gratings presented alone and odors presented alone, as described in the legend of **Supplementary Fig. 2b**. Note that responses to bimodal stimuli were not used in this classification, as the touch and odor category was defined as the cells significantly responding to both tactile gratings presented alone and odors presented alone. As a result, the proportions of cells in the touch and odor category do not match exactly with the proportion of odor-modulated cells in the presence of tactile inputs reported in **Fig. 2e** and **Fig. 4e**, which we established by comparing the responses to bimodal stimuli and tactile gratings presented alone (for the active context, 12.5% (odor only) + 5.8 (touch and odor) = 18.3% \approx 21.71% odor-modulated cells in the presence of tactile input (**Fig. 2e**); for the passive context, 3.8% (odor only) + 7% (touch and odor) = 10.8% \approx 8.99% odor-modulated cells in the presence of tactile inputs (**Fig. 2e**)). Note also that the proportions of odor only cells do match with the proportions of odor-responsive cells reported in **Fig. 2i** and **Fig. 4i** since they are established with the same statistical comparison (responses to odors presented alone compared with responses to blank stimuli). We apologize for the confusion in our previous response to this comment.

We report those proportions in the first subsection of the results section: “Among those populations, 1907 (19.9%) and 5162 (38.5%) neurons from the active and passive contexts, respectively, were responsive to at least one of the nine stimulation conditions (**Supplementary Fig. 2a**; significance threshold = 5%; Kruskal-Wallis test) and were kept for analysis. Among these populations of responsive cells, 49.4% and 68.9% responded only to touch, while 12.5% and 3.8% responded only to odors and 5.8% and 7% responded to both touch and odors in the active and passive contexts, respectively; suggesting the presence of odor-related responses (**Supplementary Fig. 2b**; significance threshold = 5%; Mann-Whitney U test).”

Supplementary figure 2. Proportion of responsive cells and distribution of response categories in the active and passive contexts. **a** Proportion of neurons responding to at least one of the nine stimulation conditions across sessions for the active context (dark) and the passive context (light). Significance threshold = 5%; Kruskal-Wallis test. Responsiveness was significantly higher in the passive context due to the sweep tactile stimulation applied to the whiskers ($p = 0.005$; Mann-Whitney U test). **b** Distribution of responsive cells responding only to tactile gratings (blue), only to odors (green) and to both tactile gratings and odors (red). A cell was classified in the touch only category if responding significantly to tactile gratings compared to blank stimuli but not responding significantly to odors (significance threshold = 5%; Mann-Whitney U test). Similarly, a cell was classified in the odor only category if responding significantly to odors but not to tactile gratings. The touch and odor category was defined as the intersection of the cells responding significantly to tactile gratings and to odors. The proportion of touch only cells was significantly higher in the passive context, while the proportion of odor only cells was lower and the proportion of cells responding to both was not significantly different ($p = 0.002$, $p = 0.002$, $p = 0.77$, respectively; Mann-Whitney U test).

My remaining comments are all relatively minor:

R2.2 Fig.6 legend needs to be updated to explain what panel c represents.

We have updated the legend.

R2.3 Fig.6f panel title and the X-axis labels appear to be mislabeled. According to the legend, the panel plots with/without touch.

We corrected the X-axis labels.

R2.4 Are the parentheses in the Fig.7f inset meant to indicate that the authors have quantitatively compared these PCs with the stated variables and there is a strong correlation? If so, I think the authors should plot this somewhere in the main or supplementary figures. If not, maybe the authors should remove the parentheses and just explicitly write in the Results that these PCs appeared to the experimenters to roughly capture X, Y, and Z. I think all we can say about Fig.7f, as is, is that the PCs do not resemble neural activity.

We have removed the parentheses on the figure. The PC weights are displayed in **Supplementary Fig. 8** (copied below) and we added to the legend what those PCs appear to capture.

Supplementary figure 8. Principal components of facial behavior. **a.** Sample frame of a video recording of facial movements made during olfactory stimulation and two-photon calcium imaging. The tube on the mouse's nose is the flow sensor for sniff monitoring. **b-d.** Weights of the three example principal components (PC) displayed in **Fig. 7f**. By visual inspection, PC1 captures global movements, PC3 captures a mode of whisker movements, and PC11 captures another mode of whisker movements as well as eyelid and jaw movements. The first 500 PCs are used for decoding of odor identity in **Fig. 7h**; using less PCs did not affect the result (**Supplementary Fig. 9**). Note that even though whisking is abolished by sectioning of the buccal and marginal mandibular branches of the facial nerve, whisker motion is still possible through their connection with facial tissue.

R2.5 Fig.7h. Is there enough data to perform decoder analysis with 500 PCs? If not, it might make sense to double check with less PCs (e.g., something like the first 10-100).

We performed the decoding analysis of odor identity with facial motion features displayed in **Fig. 7h** with only the first 100 or 10 PCs as suggested. Using less PCs did not affect our result. This analysis is provided in **Supplementary Fig. 9a, b** (copied below).

Supplementary figure 9. Odor decoding based on facial movements, respiration or whisker kinematics performs poorly independent of processing parameters. **a** Left: decoding accuracy of odor identity as in **Fig. 7h** but using the first 10 facial behavior PCs only. Chance level performance is obtained whether classification is done in single sessions (dashed line) or after pooling PCs of all sessions (plain line). Right: confusion matrix of the pooled-

session classifier. **b** Same as *b* but with the first 100 PCs. Using less PCs did not improve decoding accuracy of odor identity based on facial behavior.